# Tuning electronic structure of metal-free dual-site catalyst enables exclusive singlet oxygen production and in-situ utilization

Chao-Hai Gu [1,4], Song Wang[2,4], Ai-Yong Zhang [1,3] ✉, Chang Liu[1], Jun Jiang [2] ✉ & Han-Qing Yu [1] ✉

Developing eco-friendly catalysts for effective water purification with minimal oxidant use is imperative. Herein, we present a metal-free and nitrogen/fluorine dual-site catalyst, enhancing the selectivity and utilization of singlet oxygen ($^1O_2$) for water decontamination. Advanced theoretical simulations reveal that synergistic fluorine-nitrogen interactions modulate electron distribution and polarization, creating asymmetric surface electron configurations and electron-deficient nitrogen vacancies. These properties trigger the selective generation of $^1O_2$ from peroxymonosulfate (PMS) and improve the utilization of neighboring reactive oxygen species, facilitated by contaminant enrichment at the fluorine-carbon Lewis-acid adsorption sites. Utilizing these insights, we synthesize the catalyst through montmorillonite (MMT)-assisted pyrolysis (NFC/M). This method leverages the role of MMT as an in-situ layer-stacked template, enabling controlled decomposition of carbon, nitrogen, and fluorine precursors and resulting in a catalyst with enhanced structural adaptability, reactive site accessibility, and mass-transfer capacity. The NFC/M demonstrates an impressive 290.5-fold increase in phenol degradation efficiency than the single-site analogs, outperforming most of metal-based catalysts. This work not only underscores the potential of precise electronic and structural manipulations in catalyst design but also advances the development of efficient and sustainable solutions for water purification.

In the realm of water purification, $^1O_2$ has emerged as an exceptional candidate for selective decontamination, owing to its mild reactivity and prolonged lifespan[1–3]. Known for its high electrophilicity and specific affinity towards electron-rich substrates, $^1O_2$ minimizes the adverse effects typically associated with the formation of toxic halide byproducts[1,3,4]. Despite its benefits, the practical application of $^1O_2$ is not widespread, as it often coexists with other reactive oxygen species (ROS) within complex reaction matrices[5–8], and its effectiveness is further compromised by rapid nonradiative deactivation and mass transfer limitations in heterogeneous environments[9,10]. Moreover, metal-based catalysts pose risks of secondary pollution, while their nonmetallic counterparts often fall short in performance[11,12].

Addressing these challenges, the engineering of nanoscale adsorption-reaction dual sites on catalyst surfaces offers a promising strategy to enhance the efficacy of $^1O_2$-based processes[11,13–16]. Supported by sophisticated theoretical simulations and detailed

[1]CAS Key Laboratory of Urban Pollutant Conversion, Department of Environmental Science and Engineering, University of Science and Technology of China, Hefei, China. [2]Hefei National Research Center for Physical Sciences at the Microscale, School of Chemistry and Materials Science, University of Science and Technology of China, Hefei, China. [3]Anhui Engineering Laboratory for Rural Water Environment and Resources, School of Civil Engineering, Hefei University of Technology, Hefei, China. [4]These authors contributed equally: Chao-Hai Gu, Song Wang. ✉e-mail: ayzhang@hfut.edu.cn; jiangj1@ustc.edu.cn; hqyu@ustc.edu.cn

characterizations, this approach facilitates the precise design and controlled synthesis of high-performance catalysts[17,18]. By finely tuning the electronic configurations at these reaction sites, we can significantly boost their intrinsic reactivity and selectivity for producing specific ROS[1,19–21]. For instance, increasing the coordination number from $CuN_3$ to $CuN_4$ markedly improves site efficiency by lowering the barriers to intermediate formation and product desorption, thereby enhancing the selective production of ·OH[22]. Similarly, has been tailored to preferentially adsorb molecular oxygen, thereby facilitating the targeted generation of $^1O_2$[3]. The electron-depleted reaction site exhibited a higher adsorption selectivity for terminal O atoms in oxidant molecules and the facile $^1O_2$ generation[19,23,24]. Meanwhile, boosting pollutant affinity to the catalyst surface can shorten the migration distance and enhance the utilization efficiency of short-lived ROS. In the context of $^1O_2$-mediated water purification, strategically positioned neighboring adsorption-reaction dual sites on a single surface of a metal-free catalyst are particularly beneficial for the extensive generation and effective utilization of specific ROS through meticulous electronic manipulation.

Herein, we developed a transition metal-free dual-site catalyst designed for the selective production and effective utilization of $^1O_2$ from PMS activation. This catalyst combines nitrogen vacancies (NVs) and fluorine-carbon Lewis-acid sites as integral reaction and adsorption centers, respectively. Density function theory (DFT) calculations revealed that the electronic configuration and surface polarization were regulated by the synergistic near- and long-range interactions of nitrogen and fluorine, which optimized the adsorption of both oxidant and pollutant, as well as interfacial charge transfer. The moderate adsorption of PMS and pollutant (with phenol as a model) enabled favorable $^1O_2$ generation and utilization. Employing MMT-assisted pyrolysis, we synthesized this dual-site catalyst (NFC/M), achieving uniform co-doping of N and F. We meticulously characterized the morphology, structure, and electron properties of NFC/M, elucidating the MMT-mediated synthesis process and identifying the dual reaction centers. $^1O_2$ was confirmed as the sole ROS in the NFC/M-PMS system, with its formation mechanism meticulously detailed. Benefited from the selective generation and neighboring utilization of $^1O_2$, NFC/M exhibited exceptional performance in pollutant degradation. Our work demonstrates the rational design and controlled synthesis of metal-free dual-site catalyst to regulate the electronic structure of reactive site and the reaction mechanism of Fenton-like catalysis for safe water purification.

## Results

### Dual-site metal-free catalyst for generation and utilization of $^1O_2$

Given the notable oxygen-binding affinity and ease of formation of NVs, we chose NVs as reaction sites to design the metal-free dual-site catalyst for $^1O_2$ generation. Motivated by fluorine's exceptional ability to modulate electron behavior[25], we strategically positioned F near diverse NVs within the carbon matrix, referred to as Nv-NFC, contrasting with the baseline Nv-NC (Fig. 1a). During theoretical optimizations, the F atoms shifted slightly, yet the overall structure remained stable (Supplementary Fig. 1). DFT calculations showed significant modulation in the electronic configuration on the fluorinated catalyst's surface (Fig. 1b and Supplementary Fig. 2). F's high electronegativity and potent electron affinity led to localized charge distributions, creating distinct electron-rich and electron-deficient zones, disrupting the symmetry of surface electron distribution. This asymmetry would further benefit the adsorption of reactants and intermediates, thus contributing to the optimization of catalytic performance[26–29]. Moreover, various NV types displayed unique electronic structures (Fig. 1b and Supplementary Fig. 2), illustrating that precise electronic tuning at the active site is achievable through F and N synergy.

This synergistic electronic adjustment significantly differentiated the catalytic structures' capabilities to adsorb and activate PMS.

A schematic in Fig. 1c shows the optimal configurations for PMS activation, where the PMS adsorption energies for the three F-free catalysts were positive (Fig. 1d; 0.48, 0.93, and 0.81 eV, respectively), indicating ineffective PMS adsorption. In contrast, fluorinated catalysts showed negative adsorption energies (−3.75, −1.01, and −0.58 eV, respectively) (Fig. 1e), suggesting that introducing F to induce asymmetry is an effective strategy to enhance adsorption strength at active sites. Additionally, the varying NV types also fine-tuned the adsorption energy, with the graphitic NV showing the strongest PMS adsorption, thus creating a high energy barrier of 3.61 eV for subsequent reaction. Conversely, the catalysts with pyridinic and pyrrolic NVs presented moderate PMS adsorption energies, more conducive to ensuing reactions (Fig. 1e).

The combined influence of F and N also manifested in interfacial charge transfer dynamics. As shown in Fig. 1f, post-fluorination, a greater amount of electrons transferred from PMS to NV (0.079 $e^-$ for $N_V$-NC-2 vs 0.109 $e^-$ for $N_V$-NFC-2). Charge density differential analysis revealed that both oxygen atoms in the PMS O−O bond tended to lose electrons, shortening the bond length from 1.509 Å in free PMS to 1.476 Å and 1.469 Å in $N_V$-NC-2 and $N_V$-NFC-2, respectively (Supplementary Table 1). Meanwhile, the O−H bond length became longer (0.9970 Å in $N_V$-NC-2 and 0.9977 Å in $N_V$-NFC-2) than that in the free PMS (0.9767 Å) (Supplementary Table 1), promoting the selective breaking of the O−H bond and forming $SO_5^{\cdot-}$ (Eq. 1). By comparing the three types of NVs, the catalysts with pyrrolic N exhibited the highest charge transfer (Fig. 1f and Supplementary Fig. 3), making them energetically the most favorable in the above reaction to form $SO_5^{\cdot-}$. Once $SO_5^{\cdot-}$ was formed, it readily generated $SO_4^{2-}$ and $^1O_2$ (Supplementary Fig. 4) via the rapid self-reaction of $SO_5^{\cdot-}$ (Eq. 2)[30].

$$NVs + HSO_5^- \rightarrow SO_5^{-*} + H^+ + e^-(NVs) \qquad (1)$$

$$2e^-(NVs) + SO_5^{-*} + SO_5^{--} \rightarrow 2SO_4^{2-} + {}^1O_2 \qquad (2)$$

Furthermore, F and N's regulation of electronic structure also impacted the adsorption of phenol on the catalyst. As shown in Supplementary Figs. 5 and 6, phenol tended to be adsorbed at the F-C site. The introduction of F induced a change in the electronic structure, which significantly enhanced the phenol adsorption energy of the catalyst from −0.40 ~ −0.21 eV to −3.81 ~ −1.72 eV (Fig. 1g). The different types of NVs also provided fine tuning of the phenol adsorption energy, aligning with the trends observed in PMS adsorption (Fig. 1g). This simultaneous modulation of PMS and phenol adsorption energies underscored the synergy between the NV catalytic site and F-C adsorption site, significantly enhancing mass transfer and the utilization efficiency of $^1O_2$ (Supplementary Fig. 7).

### MMT-assisted synthesis of the dual-site catalyst

With the help of the rational design aided by theoretical simulation, we successfully synthesized a dual-site catalyst engineered for the efficient production and neighboring utilization of $^1O_2$. In this catalyst, N and F work in concert to shape the electronic framework of NVs, which, alongside F-C Lewis-acid sites, form the core of our synergistic adsorption-reaction mechanism. However, prior challenges in F-doping limited the efficient synthesis of such dual-site catalysts[31–33]. Our method leverages MMT-assisted pyrolysis (NFC/M), which we experimentally validated to ensure controllable synthesis (Fig. 2a). Also, control samples (NFC, NC/M, FC/M, and NFC/SiO₂) were prepared with some changes (details are given in "Methods"). The NFC/M catalyst showcased a unique 3D structure composed of nanosheets, greatly enhancing surface area and pore volume compared to NFC and NC/M (Fig. 2b, c and Supplementary Figs. 8–10 and Supplementary Table 2). During synthesis, MMT acted as an in-situ hard template and

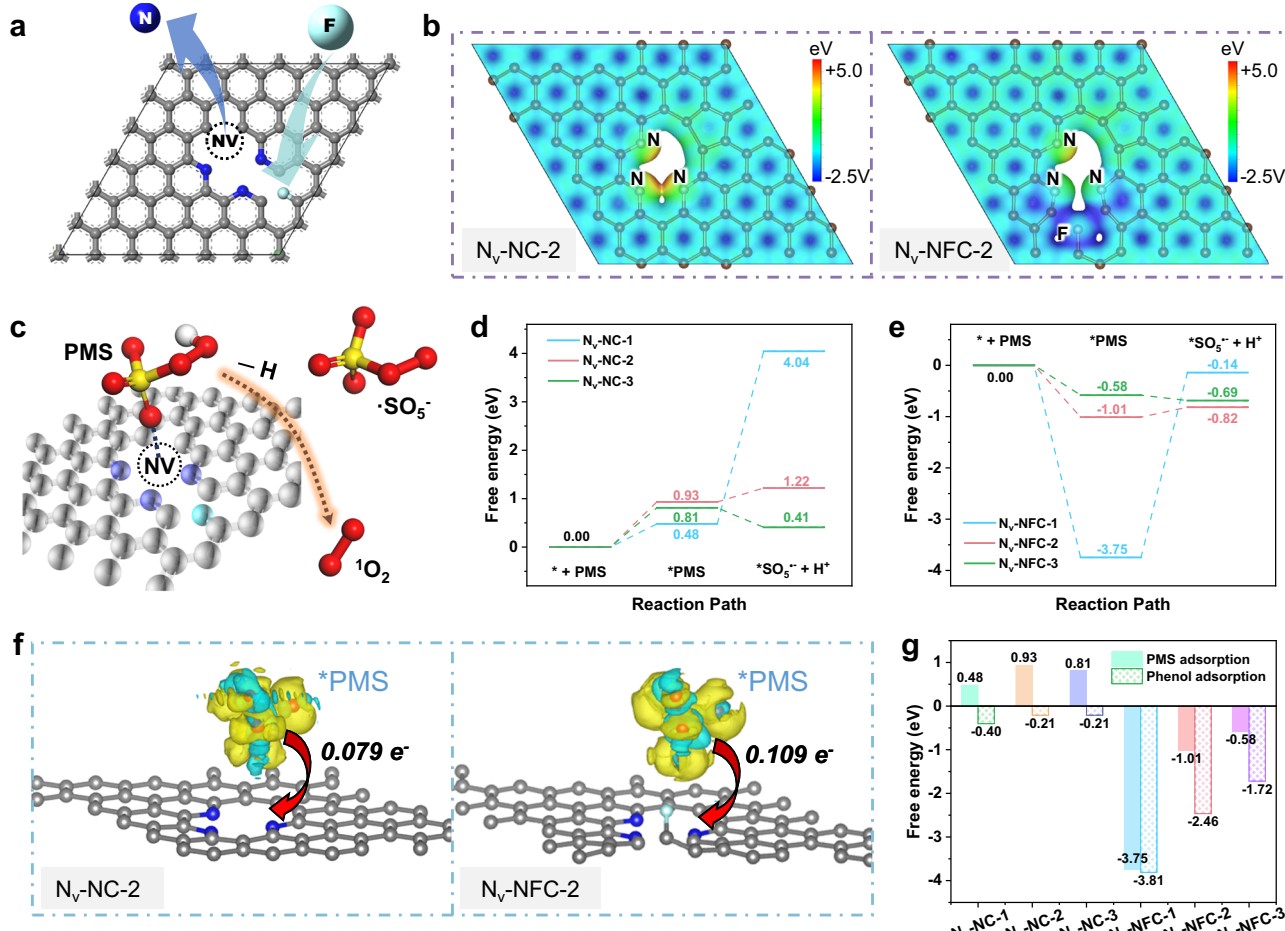

**Fig. 1 | Theoretical activity of $N_v$-NFC. a** Schematic of nitrogen vacancy and fluorine doping. **b** The electrostatic potential distributions for $N_v$-NC-2 (left) and $N_v$-NFC-2 (right). **c** Schematic representation for PMS activation, using the $N_v$-NFC-2 structure as an example. **d, e** Free energy diagram for PMS activation on different nitrogen vacancies without (**d**) and with (**e**) F doping. **f** Difference charge density for PMS adsorption on $N_v$-NC-2 (left) and $N_v$-NFC-2 (right). Yellow and cyan regions represent electron accumulation and electron depletion, respectively. Grey, blue, cyan, red, yellow, and white spheres represent C, N, F, O, S, and H atoms, respectively. **g** Free energies for PMS adsorption and phenol adsorption on different nitrogen vacancies with and without F doping. Source data are provided as a Source Data file.

was subsequently etched away by F-rich gas released from the decomposition of polytetrafluoroethylene (PTFE) (Supplementary Fig. 11 and Supplementary Table 3)[34], optimizing the catalyst's morphology and increasing the accessibility of its active sites. The resultant NFC/M consisted of randomly oriented graphitic carbon with uniform elemental distribution of C, N, and F (Fig. 2d, e and Supplementary Fig. 12b), mirroring the FC/M benchmark (Supplementary Fig. 13). In contrast, NFC synthesized without MMT exhibited a highly localized distribution of F (Supplementary Fig. 12d). These observations underscore MMT's crucial role in the controlled synthesis of the dual-site catalyst, balancing the decomposition kinetics of C, N, and F precursors during pyrolysis and catalyzing N and F co-doping.

The removal of MMT coincided with the emergence of NVs, characterized by diminished triazine units and changes in vibrational modes (Fig. 3a)[22]. The graphitic carbon structure of NFC/M stood distinct from the $C_3N_4$ structure of NC/M[35] (Supplementary Figs. 14 and 15). The characteristic D and G bands of graphite carbon appeared at 1351 and 1558 cm$^{-1}$, respectively (Fig. 3b). The $I_D/I_G$ ratio of NFC/M, standing at 1.20, surpassed those of FC/M (0.74) and NFC (1.07), indicating a higher degree of defects due to the NV formation[35]. This is further evidenced by a central shift in the D and G bands in NFC/M, suggesting localized electron distributions[36]. In addition, the increased bulk phase C/N atomic ratio of 1.47 (vs. 0.60 for NC/M and 0.91 for NFC) reaffirmed the presence of NVs in NFC/M (Fig. 3c),

aligning closely with the surface measurements (Supplementary Fig. 16a, b). Predominantly pyridinic and pyrrolic NV speciation was identified (Fig. 3d and Supplementary Fig. 17)[37]. Notably, NFC displayed a robust electron paramagnetic resonance (EPR) signal of unpaired electrons, with fewer NVs than NFC/M (Fig. 3b, c and Supplementary Fig. 16), indicating disruptions of its graphite carbon structure and increased carbon radical formation. These findings highlight the protective role of MMT during the synthesis of NFC/M. The decomposition of melamine created a reductive atmosphere, facilitating the formation of corrosive byproducts like HF (Supplementary Fig. 18), which was sequentially consumed or buffered by the $SiO_2$ and $Al_2O_3$ layers within the MMT template (Supplementary Figs. 19 and 20), effectively mitigating the corrosive byproducts from PTFE decomposition. Such a protective mechanism shields the carbon matrix from chemical etching, thus promoting the selective formation of pyridinic and pyrrolic NVs[34]. This interplay underscores the importance of MMT in safeguarding the structure of the catalyst during synthesis, ensuring the stability and functionality of the resulting NFC/M.

Abundant F-C Lewis acid sites were confirmed in NFC/M with a molar ratio of F reaching up to 4.17% (Fig. 3e and Supplementary Table 3)[35]. The strong electronegativity of F greatly enhanced the charge transfer capabilities of NFC/M, surpassing those of NFC, NC/M, and FC/M (Fig. 3f and Supplementary Figs. 21 and 22). Soft X-ray

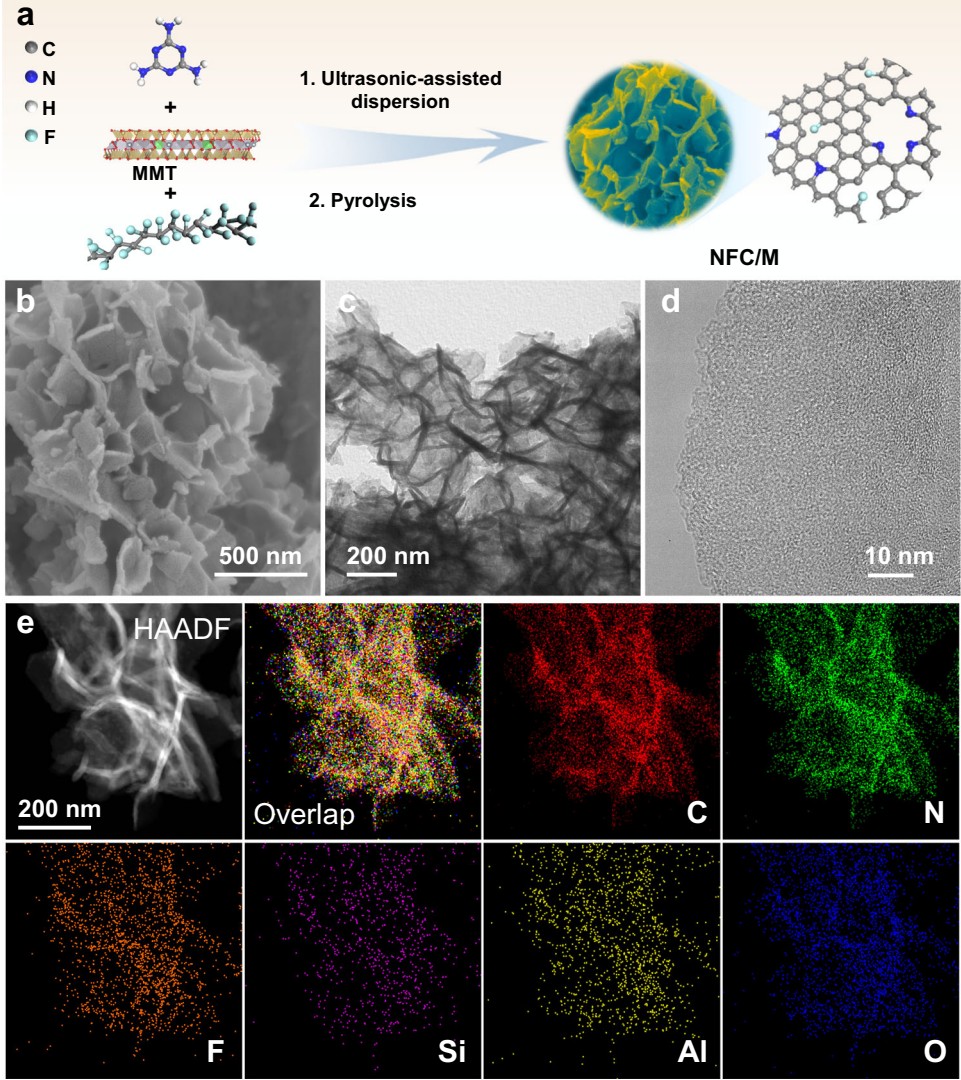

**Fig. 2 | Morphology and structure of NFC/M. a** Illustration of preparation process. **b** SEM image of NFC/M. **c** TEM image of NFC/M. **d** HRTEM image of NFC/M. **e** HAADF-STEM image and corresponding EDS elemental mapping of NFC/M.

absorption near-edge structure (XANES) analysis revealed the intricate electronic structure of NFC/M, with three distinct peaks corresponding to $\pi^*$ C−C, $\pi^*$ C−N, and $\sigma^*$ C−C/C−F orbitals evident in the C K-edge spectra of NFC/M, distinguishing it from the $C_3N_4$ signal observed in NC/M (Fig. 3g)[38]. In contrast to NFC, the intensity of the $\sigma^*$ bands of NFC/M considerably increased relative to that of the $\pi^*$ bands for C, and the $\pi^*$ C−C peak shifted toward a higher energy, indicative of increased electron localization fostered by the proliferation of C−F bonds. This shift resulted in a decreased electron density around carbon atoms[39], as confirmed by the F K-edge spectra (Fig. 3h). A significant peak at 694.8 eV denoted the C−F bond in NFC/M, its upward shift suggesting an elevated oxidation state of fluorine atoms, thereby enhancing electron density. Additionally, a prominent peak at 699.4 eV indicated F−$C_x$−N interactions among uniformly distributed F and neighboring N atoms in NFC/M, contrasting with NFC. The N K-edge showcased a notable decline in the intensity of the $\pi^*$ band relative to the $\sigma^*$ band for N in NFC/M (Fig. 3i), highlighting the prevalence of pyridinic and pyrrolic nitrogen vacancies[40,41], echoing the X-ray photoelectron spectra (XPS) analysis (Supplementary Fig. 17)[39]. These findings not only confirm the creation of pyridinic/pyrrolic NVs and F−C Lewis-acid sites through MMT-mediated pyrolysis but also demonstrate how the synergistic interplay between N's near-range effects and F's long-range influence optimally modifies their electronic

structures, crucial for the dual-site adsorption-reaction model in NFC/M.

## Exclusive $^1O_2$ production from PMS activation on NFC/M

Inspired by the well-defined geometric and electronic structure of NFC/M, the exclusive $^1O_2$ production was subsequently assessed by using multiple approaches. EPR spectroscopy reveals that using 2,2,6,6-tetramethylpiperidine (TEMP) as the $^1O_2$ trapper resulted in a notable triplet peak signal of 2,2,6,6-tetramethylpiperidine-N-oxyl radical (TEMPO). This signal greatly diminished with the introduction of NaN₃, a known $^1O_2$ quencher without impacting PMS concentration, indicating a $^1O_2$-specific reaction pathway (Fig. 4a and Supplementary Figs. 23a and 24)[1,5]. Notably, TEMPO signal intensity within the NFC/M system was substantially higher than that in the control groups and continued to rise during the reaction, showcasing its superior $^1O_2$ production capabilities[6]. When employing 5,5-dimethyl-1-pyrroline-N-oxide (DMPO) to trap potential radicals, no characteristic EPR signals indicative of ˙OH, $SO_4^{˙-}$, and $O_2^{˙-}$ were observed, but only the $^1O_2$-mediated 5,5-dimethyl-1-pyrrolidone-N-oxyl (DMPOX) signal was detected, as confirmed by the NaN₃ probe (Supplementary Fig. 23b−d)[5], further corroborating the selective generation of $^1O_2$. High-resolution mass spectrometry supports these findings by identifying the formation of DMA-$O_2$ from the

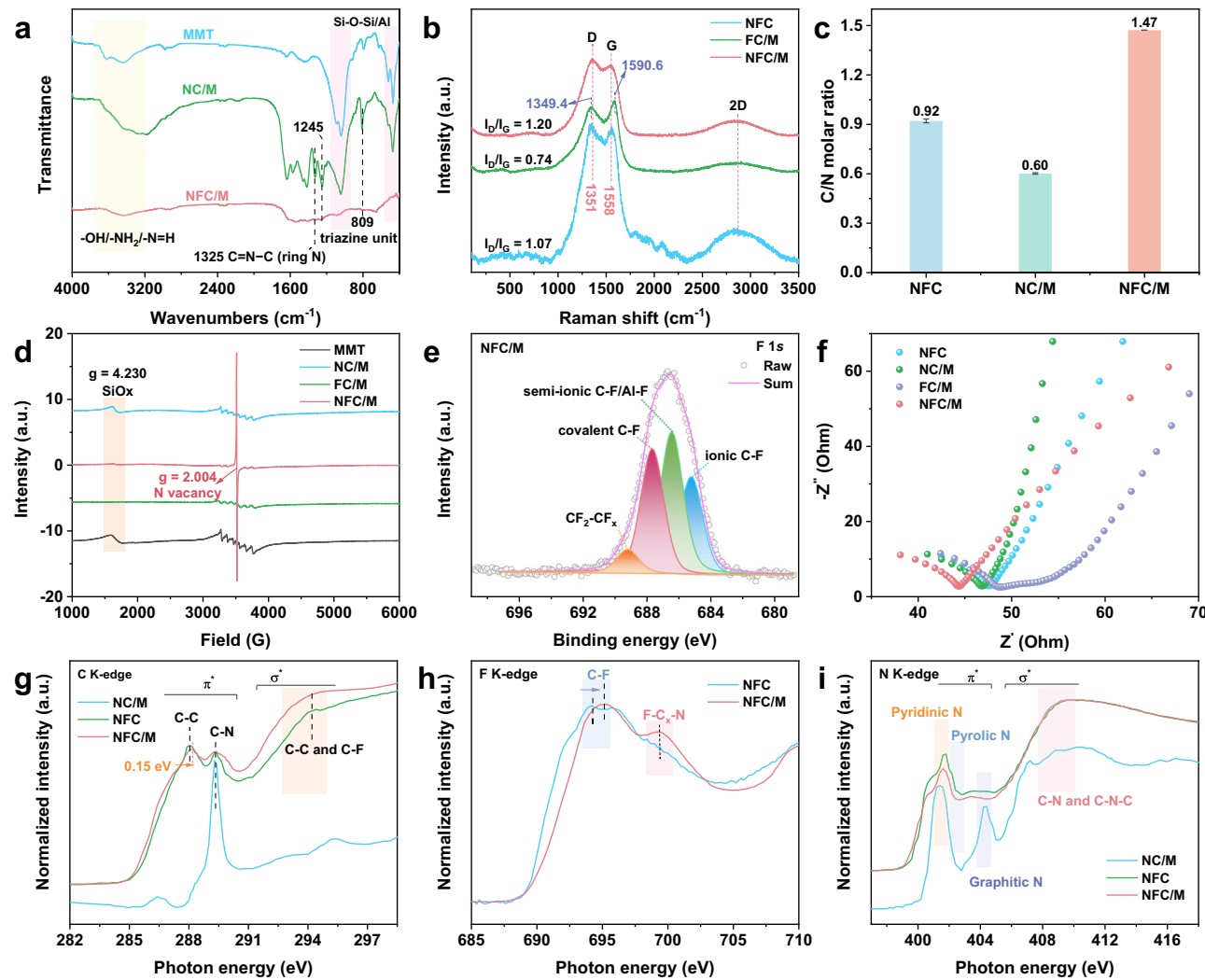

**Fig. 3 | Electronic and atomic structure of NFC/M. a** FTIR spectra. **b** Raman spectra. **c** C/N molar ratio based on elemental analysis results. Data are presented as mean values ± SD (n = 3). **d** EPR spectra measured at room temperature. **e** High-resolution XPS spectra for F 1s in NFC/M. **f** EIS spectra. **g–i** Normalized XAS spectra at the C K-edge, F K-edge, and N K-edge of different catalysts. a.u., arbitrary units. Source data are provided as a Source Data file.

interaction of 9,10-dimethylanthracene (DMA) with $^1O_2$[3,42], confirming substantial $^1O_2$ activity (Fig. 4b, c and Supplementary Figs. 25–27).

Subsequent scavenging experimental results highlight the pivotal role of $^1O_2$ in pollutant degradation (Fig. 4d). The introduction of radical-quenching agents like methanol (MeOH), ethanol (EtOH), and tert-butanol (TBA) barely influenced phenol and bisphenol A (BPA) degradation, suggesting the negligible involvement of the other reactive species (Supplementary Fig. 28a–f)[1,5]. Conversely, specific $^1O_2$-quenching agents, TEMP, NaN3, and furfuryl alcohol (FFA), at any concentration drastically inhibited pollutant degradation with the effect intensifying concomitant with the scavengers' concentration, highlighting the dominant role of $^1O_2$ in the NFC/M-mediated Fenton-like catalysis (Supplementary Fig. 28g-i). Additionally, replacing water with deuterium oxide ($D_2O$) to extend the $^1O_2$ lifespan moderately enhanced pollutant degradation, further conforming its key role in these reactions (Supplementary Fig. 29)[1]. No pollutant removal was observed in the galvanic oxidation system (GOS) with either blank or NFC/M-modified carbon electrodes, indicating negligible catalyst-mediated electron transfer from pollutant to oxidant (Supplementary Fig. 30)[43]. Therefore, $^1O_2$ was the sole active species within the NFC/M-PMS system (Fig. 4e), and responsible for the pollutant degradation.

The mechanistic insights into $^1O_2$ production were further investigated. Despite possible involvement of dissolved oxyge[14], the introduction of nitrogen or oxygen gas did not affect phenol degradation, dismissing these as sources for $^1O_2$ formation (Fig. 4f and Supplementary Fig. 31). This result confirms that PMS was the sole source of $^1O_2$. The absence of $O_2^{·-}$ detection and an increased $^1O_2$ signal in the presence of benzoquinone (BQ, a scavenger for $O_2^{·-}$) further confirm that $^1O_2$ was not derived from $O_2^{·-}$ intermediates (Fig. 4g and Supplementary Figs. 32, 33)[14,30].

In-situ Raman spectroscopy dissects the interactions between PMS molecules and the catalyst's active site. The peaks at 1060 and 884 cm$^{-1}$, corresponding to the vibrational modes of $SO_3^-$ and the O–O in PMS (H–O–O–$SO_3^-$), and a peak at 980 cm$^{-1}$ associated with the symmetric stretch of S=O bonds in $SO_4^{2-}$, demonstrated a swift conversion of PMS to $SO_4^{2-}$ following the activation by NFC/M (Fig. 4h)[44]. This transformation, along with a noticeable shift of the O–O peak to 877 cm$^{-1}$ post-interaction, suggests the decreased electron density and electron transfer from PMS to NFC/M[23]. This was supported by the consistent pollutant-independent PMS decomposition results (Supplementary Fig. 24).

Electrochemical tests further validated that PMS served as the electron donor for $^1O_2$ production. Chronoamperometric measurements showed a significant current increase upon PMS addition, and

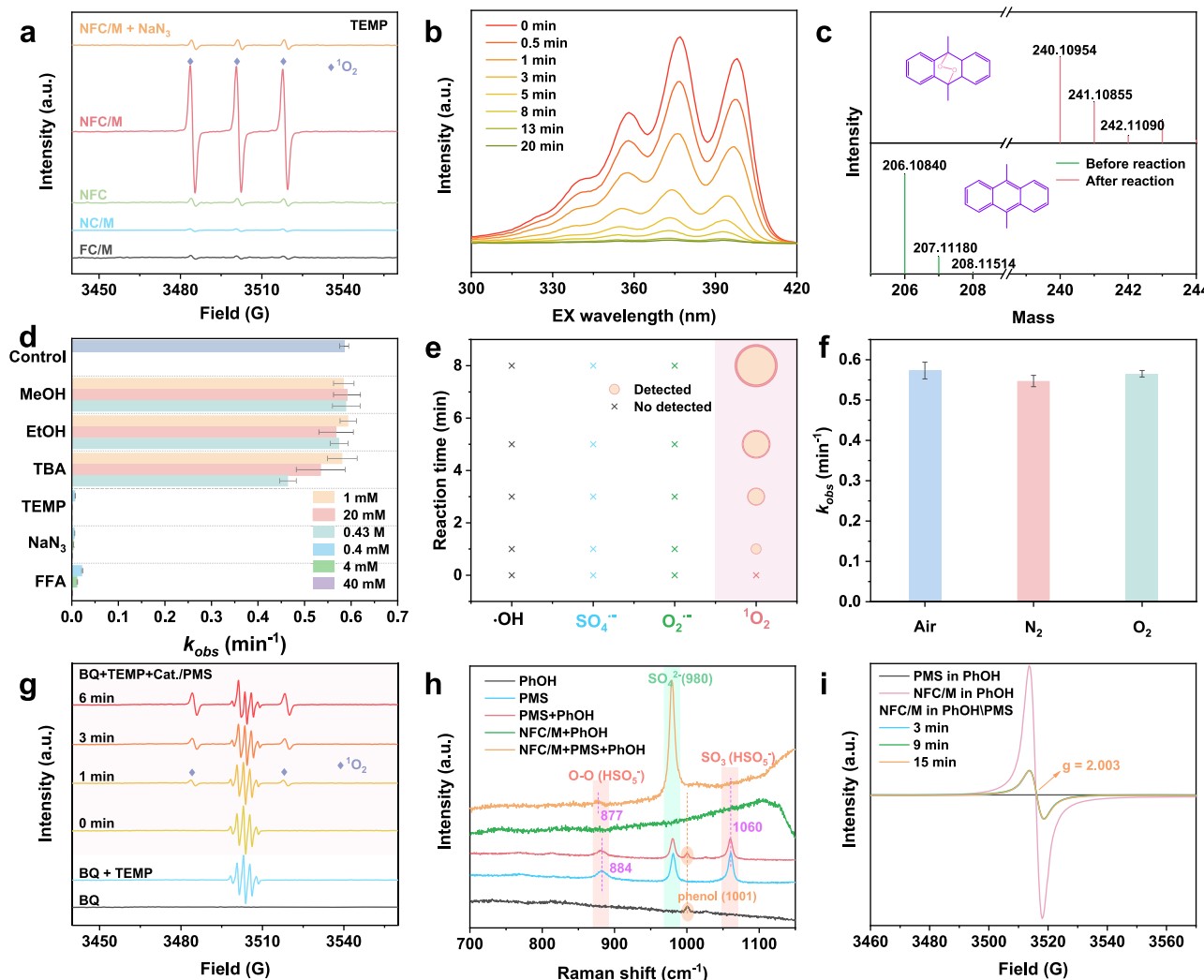

**Fig. 4 | Performance and mechanism for selective $^1O_2$ production from PMS activation on NFC/M. a** EPR spectra in different systems using TEMP as trapping agents. **b** Excitation and emission spectra of DMA in 50.0 wt.% acetonitrile in $H_2O$ from treatment by NFC/M system. The excitation was recorded from 300 to 420 nm with $\lambda_{em} = 428$ nm. **c** HR-MS chromatogram of typical DMA-$O_2$ from DMA oxidation in NFC/M system. **d** Quenching effects of the different scavengers on phenol degradation. **e** Qualitative and quantitative analyses of reactive species in NFC/M-PMS system. **f** Degradation kinetic of phenol under different atmospheres. **g** EPR spectra for the $^1O_2$ detection in the presence of TEMP and BQ. **h** In situ Raman spectra. **i** In situ EPR spectra of NFC/M reaction systems. a.u., arbitrary units. Data for (**d**) and (**f**) are presented as mean values ± SD (n = 3). Source data are provided as a Source Data file.

the open circuit potential (OCP) of the glassy carbon electrode coated with NFC/M rose immediately, indicating the strong surface interaction via electron transfer (Supplementary Fig. 34). However, phenol injection did not alter the current, signifying no electron interaction between phenol and PMS or the catalyst. A slight decrease in OCP could be attributed to the phenol adsorption on the NFC/M surface, possibly blocking active sites or modifying surface properties. Linear sweep voltammetry (LSV) analysis further confirms the electron transfer from PMS to the active sites, as evidenced by the increased current density at the NFC/M electrode (Supplementary Fig. 35)[30].

In in-situ EPR analysis, NVs were identified as key active sites. Compared to the stable unpaired electron signal on NFC without NVs (Supplementary Fig. 36), the presence of PMS led to a significant reduction in EPR signal intensity from NVs in NFC/M (Fig. 4i), indicating effective electron trapping. The signal initially decreased sharply and then stabilized, suggesting that NVs initially acted as electron acceptors to activate PMS, and subsequently as electron donors for $^1O_2$ production, quickly reaching a dynamic equilibrium (Fig. 4i and Supplementary Fig. 37). These findings, aligning with our DFT simulations,

provide robust experimental support for the mechanism of selective $^1O_2$ production through NFC/M-activated PMS (Eqs. (1) and (2)).

## Neighboring $^1O_2$ utilization on NFC/M for water decontamination

Driven by the exclusive selectivity and large production of $^1O_2$, NFC/M exhibited superior Fenton-like performance for pollutant degradation. BPA was completely degraded within 2.0 min by NFC/M-PMS system, while no BPA removal was observed within 20.0 min by PMS or NFC/M alone (Fig. 5a). Compared to control samples, NFC/M showcased remarkable reactivity towards phenol degradation, characterized by high reaction kinetics and enhanced mineralization efficiency (Fig. 5b and Supplementary Figs. 38–40). The reactivity of the Fenton-like process correlated strongly with the concentration of $^1O_2$ (Figs. 4a and 5b). In-situ Raman spectra further disclosed NFC/M's significant phenol enrichment capability, bolstering $^1O_2$ utilization (Fig. 4h). Phenol degradation followed pseudo-first-order kinetics, with NFC/M exhibiting the highest observed reaction rate constant ($k_{obs}$) of 0.58 min$^{-1}$, significantly surpassing that of NFC, NC/M, and FC/M by 58.1, 290.5, and 290.5 times, respectively (Fig. 5c).

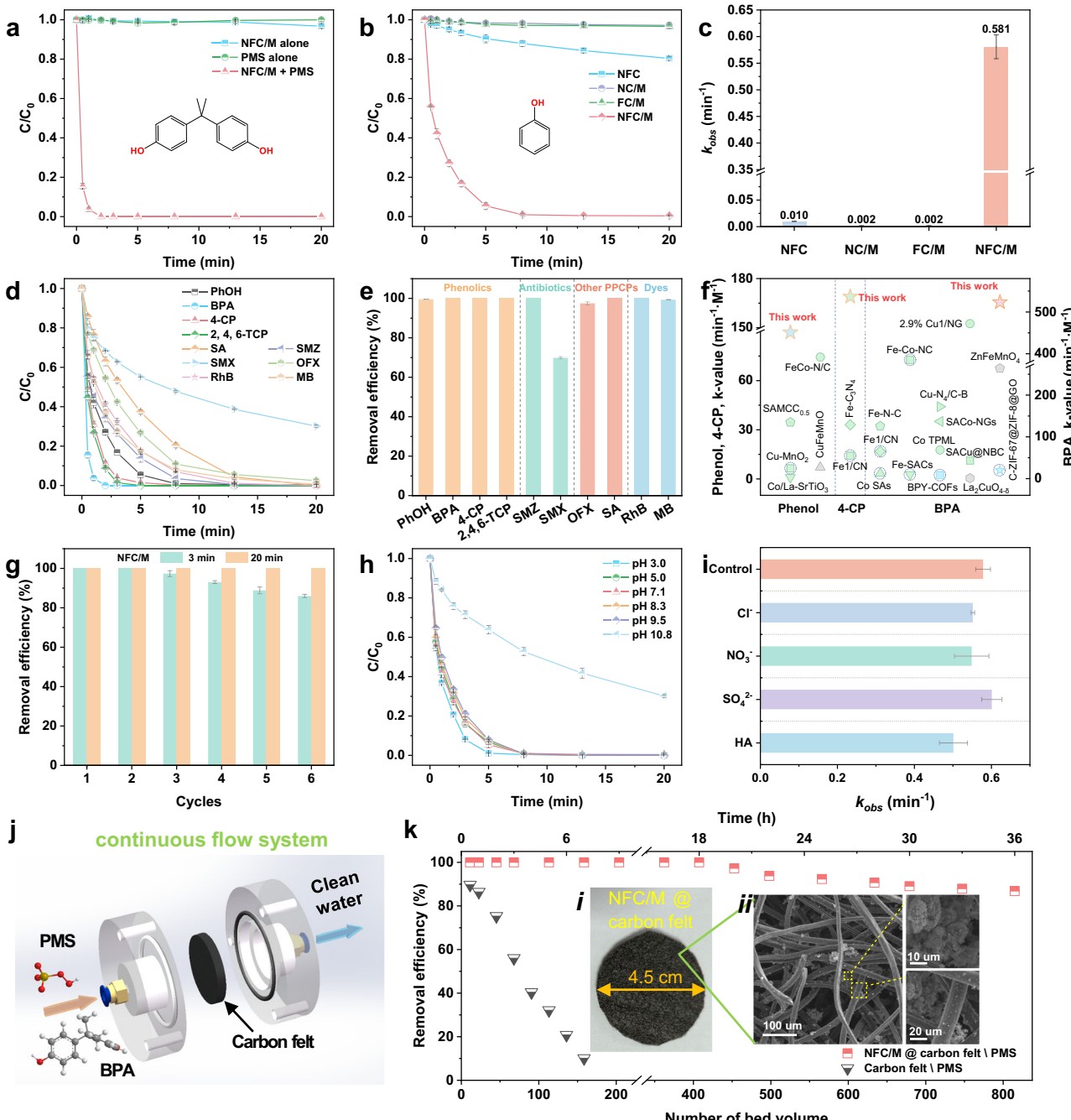

**Fig. 5 | Fenton-like performance via synergistic dual-site model. a** BPA adsorption and oxidation via PMS activation on NFC/M. **b, c** Reactivity comparison between NFC/M and references for phenol degradation. **d, e** Degradation profiles (**d**) and removal efficiency after 20 min reaction (**e**) of different pollutants in NFC/M system. **f** Comparison of degradation kinetics with recently reported PMS activation processes. The green, blue, and gray symbols represent single-atom catalysts, metal-free catalysts, and other metal-based catalysts, respectively. Among them, the symbols with purple dashed circles indicate catalysts that predominantly generate ¹O₂ as the ROS. See Supplementary Table 4 for details. **g** Reusability of NFC/M. **h** Influence of pH on phenol degradation in NFC/M system. **i** Phenol degradation kinetics with interference of ions and humic acids. **j** The schematic illustration of the continuously flow filter. **k** Continuous operation test of BPA degradation in the continuously flow filter. Inset (*i*), photograph of NFC/M @ carbon felt. Inset (*ii*), SEM images to illustrate the NFC/M catalyst coated on the carbon felt. Reaction conditions for (**a–i**): [catalyst] = 0.2 g·L⁻¹, [PMS] = 0.65 mM, [pollutant] = 20.0 mg·L⁻¹ (4-CP = 25.7 mg·L⁻¹; 2, 4, 6-TCP = 39.5 mg·L⁻¹; RhB, MB = 50.0 mg·L⁻¹), initial pH 7.0 (if not adjusted), Temp. = 20.0 ± 2.0 °C; for (**k**), flow rate = 3 mL·min⁻¹, catalyst loading = 0.1 g (if needed), [PMS] = 0.65 mM, [BPA] = 5.0 mg·L⁻¹, HRT = 2.6 min, Temp. = 20.0 ± 2.0 °C. Data for (**a–e**) and (**g–i**) are presented as mean values ± SD (n = 3). Source data are provided as a Source Data file.

Such notable disparity in reactivity also underscores that residual Si, Al, and Na within the catalyst did not contribute to its catalytic activity (Supplementary Fig. 41). Deeper insights into the catalytic activity were obtained by assessing the impacts of NVs and F content on NFC/M (Supplementary Figs. 42–45). The catalyst's performance was closely linked to these concentrations, which crucially influenced its surface electronic structure. Specifically, the catalytic activity correlated with the variations in NV concentration, initially increasing and later diminishing (Supplementary Figs. 43 and 44). This observation supports our catalyst design considerations that the electronic structure

engineering of NV drastically boosted PMS activation, resulting in excellent performance facilitated by the synergistic effects of dual-site interactions.

Further tests on the degradation of an array of nine additional pollutants validated the broad-spectrum efficacy of NFC/M (Fig. 5d). Within 20 min, various phenolic contaminants, pharmaceutical, and personal care products, as well as dyes, including p-Chlorophenol (4-CP), 2,4,6-Trichlorophenol (2,4,6-TCP), sulfamerazine (SMZ), sulfanilamide (SA), ofloxacin (OFX), rhodamine B (RhB), and methylene blue (MB), could be removed by nearly 100% (Fig. 5e). Sulfamethoxazole (SMX) was reduced by approximately 70% (Fig. 5e), while nitrobenzene (NB), a compound with electron-withdrawing properties, showed minimal degradation (Supplementary Fig. 46). These findings highlight NFC/M's selective efficacy against electron-rich pollutants, facilitated by the mild redox potential of $^1O_2$[1,45]. The performance of NFC/M surpassed that of many advanced catalysts, demonstrating the highest capacity for selective $^1O_2$ production and remarkable catalytic stability in pollutant degradation (Fig. 5f, g and Supplementary Table 4). Moreover, NFC/M exhibited considerable pH resilience, operating effectively across a range from 3.0 to 9.5, albeit with a slight performance dip at an initial pH of 10.8 due to surface repulsion and $OH^-$ quenching effects on $^1O_2$ (Fig. 5h)[46]. Additionally, phenol removal was highly resistant to interference from humic acids (HA) and various coexisting ions at all tested concentrations (Fig. 5i and Supplementary Fig. 47), further underscoring its robustness in complex water matrices.

The practical application of NFC/M in real-world scenarios was also examined, particularly in the treatment of industrial wastewater, where it achieved significant chemical oxygen demand (COD) removal efficiencies (Supplementary Tables 5 and 6). Over 50% COD removal was accomplished within 60 min in aniline production wastewater, while 48.2% of COD was removed from biochemical wastewater in the same timeframe. Considering the challenging nature of these industrial effluents, NFC/M's performance was notably impressive.

Exploring its application potential at a device level, NFC/M was integrated into a commercial carbon felt (45 mm diameter, 5 mm thickness) to construct a water-purification filter (Supplementary Fig. 48). The high surface area and porosity of carbon felt enhanced the treatment capacity of the filter[46–48]. Its effectiveness in continuous pollutant removal was assessed in a flow-through setup (Fig. 5j), where a solution containing 5 ppm BPA and 0.65 mM PMS was processed at a flow rate of 3 mL/min with a hydraulic retention time of just 2.6 min. Thanks to the abundant percolating channels and well-distributed NFC/M within the filter (Supplementary Fig. 49), it maintained over 86.8% BPA removal efficiency after 36 h of operation (815 bed volumes) (Fig. 5k). In contrast, BPA removal efficiency sharply decreased in filtration systems lacking NFC/M, primarily due to the saturated adsorption of the carbon felt. Overall, the NFC/M system showcased exceptional adaptability, durability, and catalytic performance in handling complex wastewater, affirming its viability for practical water purification scenarios.

## Discussion

Informed by DFT calculations, we devised a metal-free dual-site catalyst featuring NV reaction centers and F-C Lewis-acid adsorption centers, specifically engineered for the exclusive generation and neighboring utilization of $^1O_2$ in water purification. The catalyst's controllable synthesis was experimentally corroborated through MMT-assisted pyrolysis, capitalizing on the layered skeleton's three pivotal roles. The strategic co-doping of N and F meticulously tailored the electronic structure of NFC/M via both near- and long-range interactions, fostering an electron-deficient NV site alongside a symmetry-breaking electronic configuration that enhances reactant adsorption and charge transfer. The NV site preferentially adsorbed the termina-O of PMS to form $SO_5^{•-}$, a critical precursor for $^1O_2$, while

the F-C site effectively concentrated pollutants, thus optimizing the adjacent utilization of $^1O_2$. Underpinned by this synergistic adsorption-reaction dual-site model, NFC/M demonstrated exceptional intrinsic reactivity and cyclic stability in Fenton-like catalysis for water purification. Its efficacy was further validated across various environmental contaminants, real industrial wastewaters, and a continuous flow system, showcasing broad-spectrum effectiveness. This work not only underscores the rational design and controlled synthesis of a synergistic dual-site catalyst via electronic structure engineering but also enhances Fenton-like catalysis environmentally. It extends beyond improving reaction kinetics to include innovative catalyst design, deep mechanistic insights, and a focus on practical environmental applications and sustainability. These contributions signify a considerable advancement over the existing technologies, providing a solid foundation for developing next-generation catalysts for environmental remediation and beyond.

## Methods

### Chemicals

All chemicals used were of analytical grade and were employed without additional purification. Montmorillonite (Na-Al$_2$O$_9$Si$_3$) was sourced from Shanghai Yuanye Co., China. PTFE (60 wt.%) was purchased from Shanghai Hesen Electric Co., China. PMS (KHSO$_5$•0.5KHSO$_4$•0.5K$_2$SO$_4$, ≥42% KHSO$_5$ basis), melamine (99%), silicon dioxide (15 nm, 99%), PhOH (99.5%), BPA (99%), 4-CP (99%), 2,4,6-TCP (98%), SMX (99%), SMZ (98%), SA (99%), OFX (98%), RhB (98%), MB (98%), FFA (98%), TBA (99%), DMA (98%), BQ (99%), and TEMP (98%) were obtained from Aladdin Bio-Chem Technology Co., China. NaN$_3$ (98%) was acquired from Sigma-Aldrich. NB (99%), sodium hydroxide (NaOH, 97%), sulfuric acid (H$_2$SO$_4$, 98%), glucose (98%), MeOH (99.5%), and EtOH (99.7%) were sourced from Shanghai Chemical Reagent Co., China. DMPO (99%) was purchased from Dojindo Laboratories Co., China. All aqueous solutions were prepared using deionized water.

### Catalyst preparation

For the synthesis of NFC/M, typically, 0.5 g of montmorillonite and 0.5 g of melamine were combined in a 20.0 mL vial. To this mixture, 1.2 mL of 60 wt.% PTFE emulsion and 2.0 mL of deionized water were added and thoroughly mixed. This was followed by 30 min of ultrasonic dispersion. The vial was then sealed with aluminum foil punctured with three fine holes and heated to 550 °C at a rate of 13.0 °C/min in a muffle furnace for 1 h (Supplementary Note 2). The fluorine and nitrogen vacancy content in NFC/M could be adjusted by varying the amount of PTFE in the precursor.

For comparative analysis, control samples (NFC, NC/M, FC/M, and NFC/SiO$_2$) were also prepared. The catalyst synthesized without montmorillonite was designated as NFC, while that replacing montmorillonite with an equivalent amount of SiO$_2$ was termed as NFC/SiO$_2$. For NC/M, 0.5 g of montmorillonite and 1.0 g of melamine were placed in a vial, followed by the addition of 2.0 mL of deionized water and 30 min of ultrasonication. The sample was then pyrolyzed using the same method as NFC/M. The FC/M synthesis mirrored that of NFC/M but substituted melamine with 0.5 g of glucose.

### Catalyst characterizations

The morphology and structure of the samples were analyzed using a Quanta FEG-250 SEM (FEI Co., USA) and a ProX SEM (Phenom Co., USA), along with a transmission electron microscopy (TEM, H7700, Hitachi Co., Japan). High-resolution TEM analysis and corresponding energy-dispersive X-ray spectroscopy (EDS) mapping were conducted using a Talos F200X field-emission TEM at 200 kV. The specific surface area and pore size distribution were assessed via nitrogen adsorption-desorption isotherms using a Micromeritics ASAP 2460 system. Raman spectra were collected with a HORIBA HR800 (Japan) instrument, while EPR signals were recorded at room temperature using a ER200-

SRC spectrometer (Bruker Co., USA). Fourier-transform infrared (FTIR) spectra were acquired with a NicoletiN10 microscopy. The carbon to nitrogen (C/N) molar ratio in the catalyst samples was determined via elemental analysis (EA) with an instrument (Vario EL cube, Elementar Co., Germany), with the combustion tube heated to 950 °C and the reduction tube at 550 °C. Electrochemical impedance spectroscopy (EIS) data were collected using a CHI760E workstation. XPS were recorded on a Thermo NEXSA G2 spectrometer with an excitation source of monochromatized Al Ka ($hv$ = 1486.6 eV) and a pass energy of 40 eV. The values of binding energies were calibrated with the C 1$s$ peak of contaminant carbon at 284.80 eV. X-ray absorption spectra (XAS) for the C K-edge, N K-edge, and F K-edge were recorded at the BL12B-a beamline of the National Synchrotron Radiation Laboratory in Hefei, China.

### Catalytic tests and analytical methods

Pollutant degradation tests were conducted in a 50 mL beaker at room temperature (20.0 ± 2.0 °C) with magnetic stirring. Initially, 8.0 mg of the catalyst was added to 40.0 mL of a phenol solution (or other contaminants) at a specified concentration, followed by 2 min of ultrasonic dispersion and 20 min of magnetic stirring to ensure uniform suspension and achieve adsorption-desorption equilibrium. Subsequently, a precise amount of PMS stock solution was introduced to start the reaction. Samples were collected at regular intervals, immediately treated with L-ascorbic acid to stop the reaction, then subjected to high-speed centrifugation and filtered through a 0.22 μm PTFE filter. The analytes were quantified using ultra-performance liquid chromatography (UPLC, 1260 Infinity, Agilent Co., USA) equipped with a C18 column. All experiments were conducted in duplicate or triplicate to ensure reproducibility.

Cyclic stability was assessed through six successive BPA degradation tests. After each cycle, the catalyst was recovered by filtration, washed twice with deionized water, and resuspended in a fresh BPA solution via ultrasonication for subsequent use. The concentration of PMS was measured using low-concentration iodide methods with a UV-2450 spectrophotometer (Shimadzu Co., Japan). The effectiveness of treating real industrial wastewaters (see Supplementary Tables 5 and 6 for details) was gauged by measuring COD removal efficiency over 60 min using a DR-6000 spectrophotometer (Hach Co., USA).

### ROS identification

Quenching experiments were carried out to determine the dominant ROS in the NFC/M-PMS system using scavengers for $^1O_2$ scavengers such as $NaN_3$, TEMP, and FFA, and for ·OH and $SO_4^{·-}$ such as TBA, MeOH, and EtOH. ROS detection was conducted using spin-trapping EPR (ER200-SRC, Bruker Co., USA) at room temperature with DMPO and TEMP as spin-trapping agents. Specific experimental setups included: for the $^1O_2$ test, 1.0 mL of sample suspension was mixed with 40.0 μL of TEMP solution (50.0 g·$L^{-1}$). For the detection of ·OH and $SO_4^{·-}$, 76.0 μL of sample suspension was combined with 4.0 μL of DMPO. The $O_2^{·-}$ trapping-EPR assay was similarly conducted but a methanol medium was used.

## Data availability

The data supporting the findings of this study are available within the article and its Supplementary Information files. All other relevant raw data can be obtained from the corresponding author upon request. Source data are provided with this paper.

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

## Acknowledgements

The authors wish to thank the National Natural Science Foundation of China (52192684, 51821006, 52027815, 22076036, 22203082, 12227901), the Anhui Provincial Natural Science Foundation (2308085J23), and the Innovation Program for Quantum Science and Technology (2021ZD0303303) for supporting this work. The XAS spectra were collected at Hefei National Synchrotron Radiation Laboratory, China. The numerical calculations in this work were con-ducted on the supercomputing system in the Supercomputing Center of the University of Science and Technology of China.

## Author contributions

H. Q. Y. and C. H. G. conceived and coordinated this work. C. H. G. conducted experiments. S. W. and J. J. conducted the theoretical stu-dies. C. L. assisted with the DFT calculations. C. H. G., H. Q. Y., A. Y. Z., and S. W. analyzed the results and co-wrote the paper. All authors contributed to the discussion of manuscript.

## Competing interests

The authors declare no competing interests.
