## [Peer Review File · Nature Communications]

Tuning Electronic Structure of Metal-Free Dual-Site Catalyst Enables Exclusive Singlet Oxygen Production and In-Situ UtilizationREVIEWER COMMENTS

Reviewer #1 (Remarks to the Author):

This study described the preparation of a N/F dual-site catalyst via a synergistic electronic structure engineering strategy and demonstrated its enhanced performance of exclusive production of singlet oxygen for water purification. The authors reported that such a modification not only triggers the favorable adsorption of edge-O in PMS to selectively generate 1O_2 but also reduces its migration distance, thus enhancing the neighboring utilization of ROS on the F-C Lewis-acid adsorptive site. The mechanism analysis was provided from an atomic level and by the computational simulation. There is no doubt that the great effort has been made in this work. However, the manuscript is too redundant to convey the key points to the readership well. The investigations and conclusions are suggested to be reorganized in a more logic and concise way. Overall, this work presents a concept demonstration of efficient energy utilization to achieve the wastewater purification. The following certain points should be carefully addressed.

1. In the title, the description of “dual-site catalyst” is inappropriate and misleading, which should be instead of “metal-free dual-site catalyst” or others.
2. The abstract of the manuscript is too lengthiness is recommended to be restructured as follows keeping the words count to 200, as Nat Commun. Journal required (<https://www.nature.com/ncomms/submit/article>). 1st sentence: what was investigated (the novelty of your work); 2nd sentence: how was done (conceptualization and methodology); 3rd sentence: the essential findings (including the quantitative results); 4th sentence: the main conclusions (the breakthrough results you achieved in this study); 5th sentence: the wider significance and potential applications (to extend your findings to a broader research platform). Background sentences are not recommended to be included in the abstract part.
3. Could the authors provide more information for Supplementary Fig. 1 and Fig. 2? The coordination environment of F in Supplementary Fig. 1 seems inconsistent with that in Supplementary Fig. 2 (e.g., Supplementary Fig. 1D and Supplementary Fig. 2A; Supplementary Fig. 1F and Supplementary Fig. 2A). On the other hand, could the author provide the most favorable configurations (schematic representation for PMS activation) for Figs. 1c and d?
4. While the authors tried to use Supplementary Fig. 7 to support the effectiveness of the dual-site system, it is hard to catch the key point as the present information is quite limited. What is the meaning of “exposure”? Why the exposure in single-site system remains constant along with the distance? Why the 1O_2 (pink line) was used to be compared with two systems? It would be better to add the supplementary note near by the Figure.
5. SEM, TEM and BET characterizations were employed in this study to highlight the enhanced surface area of NFC/M. However, the BET data for NFC is missed in Supplementary Fig. 10.
6. EPR technique is generally used to investigate the intrinsic radical characters of the samples, in which the g-value is an important indicator for determining the radical species. Supplementary Fig. 16B shows that the NFC exhibited a highly enhanced EPR signal at the g value of 2.003 in comparison with NFC/M. I support the authors’ opinion that a large number of carbon radicals were produced in the framework of NFC because of the absence of MMT. However, how the authors

drawn the conclusion that the fewer nitrogen vacancies formed in NFC?

7. The information for Supplementary Figs. 20, Supplementary Fig. 21, and Fig. 4a is missed in the manuscript. The content in “Exclusive 1O_2 production from PMS activation on NFC/M” Section is very confusing and disordered, please rearrange it in a more clear way.

8. The authors announced that “the substitution of solvent H_2O with deuterium oxide (D_2O) to prolong 1O_2 lifetime notably enhanced phenol degradation (line 226-227)”. However, it is really hard to say that we can find such a significant enhancement from Supplementary Fig. 26A (kobs 0.581 min^{-1} vs. 0.660 min^{-1}).

9. Spelling mistakes: Line 115, “electrond”; Line 126, “phenol t on”; Line 309, “constructed” Line 311, “Tts”.

10. There are many manuscripts about exclusive singlet oxygen production in metal-free based Fenton-like system, what is your superiority expect for kobs?

11. The authors need to check rearrangement & formatting carefully (e.g., the references list was irregularly formatted), please rearrange appropriately.

Reviewer #2 (Remarks to the Author):

Aiming at the problem of low utilization efficiency of singlet oxygen species (1O_2) in water treatment, Gu et al. designed a transition metal-free NFC/M catalyst based on the guidance of theoretical calculation. The dual-site catalyst possessed nitrogen vacancies (NVs) and fluorine-carbon Lewis-acid sites, serving as reaction and adsorption centers, respectively. The authors proved that transition metal-free catalysts can achieve high catalytic activity guided by rational designing. The catalytic mechanism was elaborated as well. However, the manuscript contained several complexities that need be addressed before being considered for publication in Nature Communications.

1. The repeatability of NFC/M materials needs to verify. In the experimental section, it was stated that the catalyst was prepared by mixing several raw materials, and the vial was covered with a piece of aluminum foil with three fine holes. What is the purpose of this operation and how about the precision of this regulation? By this way, whether the content of F, and the N vacancy, can be precisely regulated? If possible, the correlation of their contents with the performances needs to be systematically studied.

2. Elements of the template, such as Al, Si and Na, were remained in the synthesized material. How many contents of these elements in the catalyst? Does the presence of Al, Si or Na affects the catalytic performance? How many C, N, F contents present in the catalyst?

3. The authors characterized the F-C Lewis locus and emphasized its critical role in the reaction. However, according to the preparation method, is it possible that the F elements migrate to Al and Si substrate? As Al_2O_3 and SiO_2 are very suitable for the location of active species. Also, is it possible that the F elements bind to Na due to the electronegativity?

4. The effects of concentrations of the quenching agents (TBA/methanol/ethanol) on the pollutant degradation activity should be studied and discussed.

5. The author should give more deeply discussion towarding the identification of the free radicals, especially the hydroxyl radicals and the sulfate radicals, and the possibility of electron transfer during the process.
6. In Fig 3c, the author used SEM-EDS to obtain the C/N ratio of the catalyst. More reliable data are suggested to provid.
7. The abstract is too long. The author is suggested to refine the language.

Response to Reviewer 1's comments

This study described the preparation of a N/F dual-site catalyst via a synergistic electronic structure engineering strategy and demonstrated its enhanced performance of exclusive production of singlet oxygen for water purification. The authors reported that such a modification not only triggers the favorable adsorption of edge-O in PMS to selectively generate $^1\text{O}_2$ but also reduces its migration distance, thus enhancing the neighboring utilization of ROS on the F-C Lewis-acid adsorptive site. The mechanism analysis was provided from an atomic level and by the computational simulation. There is no doubt that the great effort has been made in this work. However, the manuscript is too redundant to convey the key points to the readership well. The investigations and conclusions are suggested to be reorganized in a more logic and concise way. Overall, this work presents a concept demonstration of efficient energy utilization to achieve the wastewater purification. The following certain points should be carefully addressed.

The reviewer's valuable time and efforts on our manuscript are greatly appreciated.

1. In the title, the description of “dual-site catalyst” is inappropriate and misleading, which should be instead of “metal-free dual-site catalyst” or others.

Corrected as “*Tuning Electronic Structure of Metal-Free Dual-Site Catalyst Enables Exclusive Singlet Oxygen Production and In-Situ Utilization*”, as suggested.

2. The abstract of the manuscript is too lengthiness is recommended to be restructured as follows keeping the words count to 200, as *Nat. Commun.* Journal required (<https://www.nature.com/ncomms/submit/article>). 1st sentence: what was investigated (the novelty of your work); 2nd sentence: how was done (conceptualization and methodology); 3rd sentence: the essential findings (including the quantitative results); 4th sentence: the main conclusions (the breakthrough results you achieved in this study); 5th sentence: the wider significance and potential applications (to extend your findings to a broader research platform). Background sentences are not recommended to be included in the abstract part.

Accepting the reviewer's suggestion, we have rewritten the abstract as follows (P2, L1-19):

Developing eco-friendly catalysts for effective water purification with minimal oxidant use is imperative. Herein, we present a novel metal-free and nitrogen/fluorine dual-site catalyst, enhancing the selectivity and utilization of singlet oxygen ($^1\text{O}_2$) for water decontamination. Advanced theoretical simulations reveal that synergistic fluorine-nitrogen interactions modulate electron distribution and polarization, creating asymmetric surface electron configurations and electron-deficient nitrogen vacancies. These properties trigger the selective generation of $^1\text{O}_2$ from peroxymonosulfate (PMS) and improve the neighboring utilization of reactive oxygen species, facilitated by contaminant enrichment at the fluorine-carbon Lewis-acid adsorption sites. Utilizing these insights, we synthesize the catalyst through montmorillonite (MMT)-assisted pyrolysis (NFC/M). This method leverages the role of MMT as an in-situ layer-stacked

template, enabling controlled decomposition of carbon, nitrogen, and fluorine precursors and resulting in a catalyst with enhanced structural adaptability, reactive site accessibility, and mass-transfer capacity. The NFC/M demonstrates an impressive 290.5-fold increase in phenol degradation efficiency than the single-site analogs, outperforming most of metal-based catalysts. This work not only underscores the potential of precise electronic and structural manipulations in catalyst design, but also advances the development of efficient and sustainable solutions for water purification.

3. Could the authors provide more information for Supplementary Fig. 1 and Fig. 2? The coordination environment of F in Supplementary Fig. 1 seems inconsistent with that in Supplementary Fig. 2 (e.g., Supplementary Fig. 1D and Supplementary Fig. 2A; Supplementary Fig. 1F and Supplementary Fig. 2A). On the other hand, could the author provide the most favorable configurations (schematic representation for PMS activation) for Figs. 1c and d?

Accepting the reviewer's suggestion, we have provided more information about the catalyst structure models and the most favorable configuration for PMS activation into the revised manuscript (Figs. R1-R4).

Fig. R1 | Configurations of different N vacancy atomic models with and without F

doping after DFT optimization. **(a, d)** N_V -NC-1 and N_V -NFC-1, graphitic nitrogen vacancy. **(b, e)** N_V -NC-2 and N_V -NFC-2, pyridinic nitrogen vacancy. **(c, f)** N_V -NC-3 and N_V -NFC-3, pyrrolic nitrogen vacancy. Grey, blue, and cyan spheres represent C, N and F atoms, respectively.

In our previous manuscript, the inconsistency between **Supplementary Fig. 1** and **Fig. 2** arose because **Supplementary Fig. 1** depicts pre-optimization catalyst structures, while **Supplementary Fig. 2** shows the post-optimization structures. During the theoretical optimization process, although the overall structures remained stable, there were slight shifts in the positions of the F atoms, causing the changes of their coordination environment.

In the revised *Supplementary Information*, we have updated all the structures in **Supplementary Fig. 1** with the optimized structures (**Fig. R1**) and added element labels for F and N atoms in **Fig. 1b** and **Supplementary Fig. 2** (**Figs. R2** and **R3**). Moreover, we have also provided a schematic representation for PMS activation to present the most favorable configurations (**Fig. R4**).

Fig. R2 | The electrostatic potential distributions for N_V -NC-2 (left) and N_V -NFC-2 (right).

Fig. R3 | Electrostatic potential distributions for **(a)** N_V -NFC-1, **(b)** N_V -NFC-3, **(c)** N_V -NC-1 and **(d)** N_V -NC-3.

Fig. R4 | Schematic representation for PMS activation with the N_V -NFC-2 structure as an example.

To address the reviewer's concern, we have added these new theoretical results into the revised manuscript (Fig. 1b, c; Supplementary Fig. 1; Supplementary Fig. 2).

During theoretical optimizations, the F atoms shifted slightly, yet the overall structure remained stable (Supplementary Fig. 1). (P5, L78-80)

A schematic in Fig. 1c shows the optimal configurations for PMS activation, where the PMS adsorption energies for the three F-free catalysts were positive (Fig. 1d; 0.48, 0.93 and 0.81 eV, respectively), indicating ineffective PMS adsorption. (P6, L90-93)

4. While the authors tried to use Supplementary Fig. 7 to support the effectiveness of the dual-site system, it is hard to catch the key point as the present information is quite limited. What is the meaning of "exposure"? Why the exposure in single-site system remains constant along with the distance? Why the 1O_2 (pink line) was used to be compared with two systems? It would be better to add the supplementary note near by the Figure.

We highly appreciate the reviewer's constructive feedback. In our previous schematic representation (Supplementary Fig. 7), "exposure" specifically denotes the levels of reactive oxygen species (ROS) and organic pollutants at varying distances from the catalyst surface (dual-site catalyst N_V -NFC or single-site catalyst N_V -NC), and their accessibility to each other during Fenton-like reactions.

For the spatial distribution of 1O_2 , its concentration decreased rapidly with the increasing distance from the catalyst surface due to its short lifetime in an aqueous medium (2.9 ~ 4.6 μ s) (Kang et al., *Nat. Commun.* 2023, 14, 6590; Kai et al., *Environ. Sci. Technol.* 2023, 57, 7568-7577; Britt et al., *Environ. Sci. Technol.* 2012, 46, 7222-7229). Such a decline is primarily due to the nonradiative relaxation of 1O_2 , leading to rapid physical deactivation and further compounded by mass transfer limitations inherent in heterogeneous reactions (Liu et al., *Nat. Commun.* 2023, 14, 2881; Zhang et al., *Environ. Sci. Technol.* 2020, 54, 10868-10875).

As for the distribution of pollutants, the systems with a single reactive site (N_V -NC) exhibited a uniform distribution throughout the solution, as there were no adsorptive sites on the catalyst to concentrate the pollutants (Fig. 1g). This resulted in inefficient utilization of 1O_2 in Fenton-like catalysis (Zhang et al., *Environ. Sci. Technol.* 2020, 54, 10868-10875; Li et al., *J. Am. Chem. Soc.* 2018, 140, 12469-12475). In

contrast, in the metal-free dual-site system (N_v -NFC), the F-C Lewis acid moiety functioned as adsorptive sites that effectively attracted and concentrated pollutants at the catalyst surface. This configuration reduced the migration distance and increased the spatial accessibility of reactive oxygen species (the non-radical 1O_2 in our study) for the neighboring utilization with a high efficiency.

In this schematic, 1O_2 was chosen as the primary non-radical ROS to compare single-site and dual-site reaction systems in heterogeneous Fenton-like catalysis (Supplementary Fig. 7). This selection is supported by the exclusive generation of 1O_2 at engineered nitrogen vacancy sites (NVs) from PMS activation, and its effective utilization at F-C Lewis sites for pollutant degradation, as confirmed by both theoretical simulations and experimental results in NC/M and NFC/M systems (Figs. 1, 4 and Supplementary Figs. 23-29). Similarly, it is reasonable that the metal-free dual-site reaction system could be also viable for other reactive oxygen species with both high reactivity and short lifetime in the heterogeneous Fenton-like catalysis for oxidant activation and pollutant degradation.

To address the reviewer's concern, we have further elaborated on these points in the revised manuscript by replacing "exposure" with "concentration" to more precisely describe the essential dynamics between ROS and pollutants (Supplementary Fig. 7).

This simultaneous modulation of PMS and phenol adsorption energies underscored the synergy between the NV catalytic site and F-C adsorption site, significantly enhancing mass transfer and the utilization efficiency of 1O_2 (Supplementary Fig. 7). (P7, L121-124)

Notes for Supplementary Fig. 7

The term "concentration" specifically denotes the level of reactive oxygen species (here is 1O_2) and organic pollutants at varying distances from the catalyst surface (dual-site catalyst N_v -NFC or single-site catalyst N_v -NC) and their accessibility to each other during Fenton-like reactions.

For the spatial distribution of 1O_2 , its concentration rapidly decreased with the increasing distance from the catalyst surface due to its brief lifetime in an aqueous medium ($2.9 \sim 4.6 \mu s$)⁸⁻¹⁰. This decline was primarily due to the nonradiative relaxation of 1O_2 , leading to the rapid physical deactivation and further compounded by mass transfer limitations inherent in heterogeneous reactions^{11, 12}.

As for the distribution of pollutants, the systems with a single reactive site (N_v -NC) exhibited a uniform distribution throughout the solution, as there were no adsorptive sites on the catalyst to concentrate the pollutants (Fig. 1g). This resulted in inefficient utilization of 1O_2 during Fenton-like catalysis^{12, 13}. In contrast, in the metal-free dual-site system (N_v -NFC), the F-C Lewis acid sites functioned as adsorptive sites, which effectively attracted and concentrated pollutants at the catalyst surface. This configuration enhanced the proximity and availability of reactive species, significantly improving the in-situ utilization of 1O_2 . Due to the exclusive production of 1O_2 at the nitrogen vacancy sites (NVs) triggered by PMS activation and its neighboring utilization at the F-C Lewis sites for pollutant degradation, 1O_2 was utilized as the non-radical reactive oxygen species for comparison between single-site and dual-site

systems in Fenton-like catalysis (Figs. 1 and 4). The dual-site system also supported other reactive oxygen species with high reactivity and short lifetimes in heterogeneous Fenton-like processes.

Supplementary References

8. Kang, Y. et al. Unveiling the spatially confined oxidation processes in reactive electrochemical membranes. *Nat. Commun.* **14**, 6590 (2023).
9. Cheng, K., Zhang, L. & McKay, G. Evaluating the microheterogeneous distribution of photochemically generated singlet oxygen using furfuryl amine. *Environ. Sci. Technol.* **57**, 7568-7577 (2023).
10. Lenzen, M., Kanemoto, K., Moran, D. & Geschke, A. Mapping the structure of the world economy. *Environ. Sci. Technol.* **46**, 8374-8381 (2012).
11. Liu, T. et al. Water decontamination via nonradical process by nanoconfined Fenton-like catalysts. *Nat. Commun.* **14**, 2881 (2023).
12. Zhang, S. et al. Mechanism of heterogeneous fenton reaction kinetics enhancement under nanoscale spatial confinement. *Environ. Sci. Technol.* **54**, 10868-10875 (2020).
13. Li, X. et al. Single cobalt atoms anchored on porous n-doped graphene with dual reaction sites for efficient fenton-like catalysis. *J. Am. Chem. Soc.* **140**, 12469-12475 (2018).

5. SEM, TEM and BET characterizations were employed in this study to highlight the enhanced surface area of NFC/M. However, the BET data for NFC is missed in Supplementary Fig. 10.

Provided (Supplementary Fig. 10; Supplementary Table 2), as suggested.

The BET analysis shows a largely lower specific surface area for NFC (29.61 m²/g) compared to the enhanced 69.72 m²/g of NFC/M (**Fig. R5** and **Table R1**). This difference underscored the effective nanosheet-assembled 3D structure of NFC/M, attributed to the layer-stacked MMT template used during synthesis.

Fig. R5 | Specific surface area. **(a)** N₂ adsorption and desorption isotherms, **(b)** adsorption-pore size distribution, and **(c)** desorption-pore size distribution for NFC/M, NC/M and NFC.

Table R1 | Summary of N₂ sorption for NFC/M, NC/M and NFC.

Catalyst	BET Surface Area (m ² /g)	Pore Volume (cm ³ /g)
NFC/M	69.72	0.27
NC/M	14.66	0.06
NFC	29.61	0.12

To address the reviewer's concern, we have added these new results and descriptions into the revised manuscript (Supplementary Fig. 10; Supplementary Table

2).

The NFC/M catalyst showcased a unique 3D structure composed of nanosheets, greatly enhancing surface area and pore volume compared to NFC and NC/M (Fig. 2b, c and Supplementary Figs. 8-10 and Supplementary Table 2). (P8, L136-138)

6. EPR technique is generally used to investigate the intrinsic radical characters of the samples, in which the g-value is an important indicator for determining the radical species. Supplementary Fig. 16B shows that the NFC exhibited a highly enhanced EPR signal at the g value of 2.003 in comparison with NFC/M. I support the authors' opinion that a large number of carbon radicals were produced in the framework of NFC because of the absence of MMT. However, how the authors drawn the conclusion that the fewer nitrogen vacancies formed in NFC?

We appreciate the reviewer's insightful comments regarding the use of the EPR technique and the interpretation of the g-value as an indicator of radical species. Indeed, the g-value of ~2.003 observed in the NFC sample could signify the presence of persistent carbon-centric free radicals with adjacent oxygen atoms, or nitrogen vacancies causing unpaired electrons. However, distinguishing among these requires an analysis of the structural context of each sample (Arianna et al., *Angew. Chem. Int. Ed.* 2022, 61, e202210640; Can et al., *Angew. Chem. Int. Ed.* 2017, 56, 6627-6631).

In response to the reviewer's query about the formation of nitrogen vacancies, we have conducted a comprehensive analysis combining both Raman spectroscopy and elemental analysis. First, Raman spectroscopy revealed the characteristic D and G bands of graphite carbon at 1351 and 1558 cm^{-1} , respectively, for both NFC and NFC/M (Fig. 3b). Notably, the I_D/I_G ratio for NFC/M was 1.20, greatly higher than NFC's 1.07 (Fig. 3b), suggesting fewer structural defects in NFC (Liu et al., *Adv. Mater.* 2020, 32, 1907690).

Fig. R6 | Elemental analysis and structural defect. **(a)** Comparative analysis of the C/N molar ratio on the catalyst surface based on SEM-EDS results. **(b)** Comparison of C/N molar ratios in catalyst bulk phases using elemental analysis performed on an instrument (Vario EL cube, Elementar Co., Germany). **(c)** EPR spectra of NFC/M and NFC measured at room temperature.

Additionally, the XPS and EDS analyses provided further clarity. Despite the equal mass ratios of carbon, nitrogen, and fluorine precursors for both NFC and NFC/M, NFC exhibited a notably lower surface C/N ratio compared to NFC/M, as evidenced by XPS and EDS results (Supplementary Fig. 16a, b). This suggests minimal nitrogen loss in NFC's synthesis without the MMT template, resulting in fewer nitrogen vacancies on the surface and sub-surface of its carbon matrix (Su et al., *Angew. Chem. Int. Ed.* 2021, 133, 21431-21436). To confirm these findings, we have conducted detailed elemental

analyses of the bulk phase using a precise elemental analysis (EA) test (**Fig. R6b**). The results affirmed that the C/N molar ratio in NFC was substantially lower than that in NFC/M, indicating the reduced nitrogen vacancy formation in NFC (Kumar et al., *J. Mater. Chem. A*, 2021, 9, 111). These analyses align closely with the surface measurements, providing a comprehensive understanding of the elemental composition and vacancy distribution in the catalysts.

Hence, the EPR signal at $g = 2.003$ in NFC strongly suggests a high level of unpaired electrons (**Fig. R6c**), likely due to the corrosive effects of fluorine precursor decomposition and subsequent carbon radical formation. The comprehensive structural and compositional analyses support a fewer incidence of nitrogen vacancies in NFC compared to NFC/M.

To address the reviewer's concern, we have updated these additional data and interpretations in the revised manuscript (**Fig. 3c**; **Supplementary Fig. 16**).

The characteristic D and G bands of graphite carbon appeared at 1351 and 1558 cm^{-1} , respectively (**Fig. 3b**). The I_D/I_G ratio of NFC/M, standing at 1.20 , surpassed those of FC/M (0.74) and NFC (1.07), indicating a higher degree of defects due to the NV formation³⁵. (**P9, L153-156**)

In addition, the increased bulk phase C/N atomic ratio of 1.47 (vs. 0.60 for NC/M and 0.91 for NFC) reaffirmed the presence of NVs in NFC/M (**Fig. 3c**), aligning closely with the surface measurements (**Supplementary Fig. 16a, b**). (**P9, L157-160**)

Notably, NFC displayed a robust EPR signal of unpaired electrons, with fewer NVs than NFC/M (**Fig. 3b, c** and **Supplementary Fig. 16**), indicating disruptions of its graphite carbon structure and increased carbon radical formation. (**P9, L161-164**)

The carbon to nitrogen (C/N) molar ratio in the catalyst samples was determined via elemental analysis (EA) with an instrument (Vario EL cube, Elementar Co., Germany), with the combustion tube heated to 950 °C and the reduction tube at 550 °C. (**P20, L409-412**)

Supplementary Fig. 16 | Elemental analysis and structural defect. (a) The C/N molar ratio based on XPS results. (b) The C/N molar ratio based on SEM-EDS results. (c) EPR spectra of NFC/M and NFC measured at room temperature.

Notes for Supplementary Fig. 16

The composite EPR signal at $g=2.003$ indicated the presence of either persistent carbon-centric free radicals or defective nitrogen vacancies. Despite equal precursor ratios for carbon, nitrogen, and fluorine, NFC showed a substantially lower surface C/N ratio than NFC/M, as detailed in the XPS and EDS results (Supplementary Fig. 16a, b). Such a lower ratio suggested minimal nitrogen loss during the synthesis of NFC, which lacked the protective MMT template, resulting in fewer nitrogen vacancies in the carbon matrix. This finding was corroborated by bulk phase elemental analysis (EA), which aligns with surface measurements and provides a detailed view of the catalysts' elemental composition and vacancy distribution (Fig. 3c).

Moreover, Raman spectroscopic results (Fig. 3b) support the conclusion that fewer nitrogen vacancies were present in NFC. However, the EPR characterization (Supplementary Fig. 16c) revealed a significantly strong unpaired electron signal—11.2 times higher than that in NFC/M—highlighting the absence of MMT's buffering effect. The lack of this protection allowed the corrosive gases from fluorine precursor decomposition to etch and structurally damage the NFC surface, leading to an increased formation of carbon radicals.

7. The information for Supplementary Figs. 20, Supplementary Fig. 21, and Fig. 4a is missed in the manuscript. The content in “Exclusive $^1\text{O}_2$ production from PMS activation on NFC/M” Section is very confusing and disordered, please rearrange it in a more clear way.

We apologize for the confusion caused by the arrangement in our previous submission and acknowledge that it diverges from the conventional structure typically found in literature, where scavenging tests and other analyses typically follow discussion on environmental Fenton-like catalysis to elucidate reaction mechanisms.

In our work, the sequence was specifically designed to emphasize the direct evidence to support exclusive $^1\text{O}_2$ production from PMS activation on NFC/M. This includes the results from chemical scavenging tests, electron paramagnetic resonance (EPR) measurements, Raman spectroscopy, liquid chromatography-mass spectrometry (LC-MS), and electrochemical characterizations. These methods were applied not merely to probe the reaction mechanisms and pollutant degradation—as is common—but to substantiate the unique behaviors of our catalyst system.

To address the reviewer's concerns, we have revised this section to provide clearer explanations and reorganized the content to improve readability and logical flow. (P11-14, L199-270)

Exclusive $^1\text{O}_2$ production from PMS activation on NFC/M

Inspired by the well-defined geometric and electronic structure of NFC/M, the exclusive $^1\text{O}_2$ production was subsequently assessed by using multiple approaches. EPR spectroscopy reveals that using 2,2,6,6-tetramethylpiperidine (TEMP) as the $^1\text{O}_2$

trapper resulted in a notable triplet peak signal of 2,2,6,6-tetramethylpiperidine-N-oxyl radical (TEMPO). This signal greatly diminished with the introduction of NaN₃, a known ¹O₂ quencher without impacting PMS concentration, indicating a ¹O₂-specific reaction pathway (Fig. 4a and Supplementary Figs. 23a and 24)¹⁻⁵. Notably, TEMPO signal intensity within the NFC/M system was substantially higher than that in the control groups and continued to rise during the reaction, showcasing its superior ¹O₂ production capabilities⁶. When employing 5,5-dimethyl-1-pyrroline-N-oxide (DMPO) to trap potential radicals, no characteristic EPR signals indicative of •OH, SO₄^{•-}, and O₂^{•-} were observed, but only the ¹O₂-mediated DMPOX signal was detected, as confirmed by the NaN₃ probe (Supplementary Fig. 23b-d)⁵, further corroborating the selective generation of ¹O₂. High-resolution mass spectrometry supports these findings by identifying the formation of DMA-O₂ from the interaction of 9,10-dimethylantracene (DMA) with ¹O₂^{3,42}, confirming substantial ¹O₂ activity (Fig. 4b-c and Supplementary Figs. 25-27).

Subsequent scavenging experimental results highlight the pivotal role of ¹O₂ in pollutant degradation (Fig. 4d). The introduction of radical-quenching agents like methanol (MeOH), ethanol (EtOH), and tert-butanol (TBA) barely influenced phenol and bisphenol A (BPA) degradation, suggesting the negligible involvement of the other reactive species (Supplementary Fig. 28a-f)^{1,5}. Conversely, specific ¹O₂-quenching agents, TEMP, NaN₃ and furfuryl alcohol (FFA), at any concentration drastically inhibited pollutant degradation with the effect intensifying concomitant with the scavengers' concentration, highlighting the dominant role of ¹O₂ in the NFC/M-mediated Fenton-like catalysis (Supplementary Fig. 28g-i). Additionally, replacing water with deuterium oxide (D₂O) to extend the ¹O₂ lifespan moderately enhanced pollutant degradation, further conforming its key role in these reactions (Supplementary Fig. 29)¹. No pollutant removal was observed in the galvanic oxidation system (GOS) with either blank or NFC/M-modified carbon electrodes, indicating negligible catalyst-mediated electron transfer from pollutant to oxidant (Supplementary Fig. 30)⁴³. Therefore, ¹O₂ was the sole active species within the NFC/M-PMS system (Fig. 4e), and responsible for the pollutant degradation.

The mechanistic insights into ¹O₂ production were further investigated. Despite possible involvement of dissolved oxygen¹⁴, the introduction of nitrogen or oxygen gas did not affect phenol degradation, dismissing these as sources for ¹O₂ formation (Fig. 4f and Supplementary Fig. 31). This result confirms that PMS was the sole source of ¹O₂. The absence of O₂^{•-} detection and an increased ¹O₂ signal in the presence of benzoquinone (BQ, a scavenger for O₂^{•-}) further confirm that ¹O₂ was not derived from O₂^{•-} intermediates (Fig. 4g and Supplementary Figs. 32, 33)^{14,30}.

In-situ Raman spectroscopy dissects the interactions between PMS molecules and the catalyst's active site. The peaks at 1060 and 884 cm⁻¹, corresponding to the vibrational modes of SO₃⁻ and the O-O in PMS (H-O-O-SO₃⁻), and a peak at 980 cm⁻¹ associated with the symmetric stretch of S=O bonds in SO₄²⁻, demonstrated a swift conversion of PMS to SO₄²⁻ following the activation by NFC/M (Fig. 4h)⁴⁴. This transformation, along with a noticeable shift of the O-O peak to 877 cm⁻¹ post-interaction, suggests the decreased electron density and electron transfer from PMS to

NFC/M²³. This was supported by the consistent pollutant-independent PMS decomposition results (Supplementary Fig. 24).

Electrochemical tests further validated that PMS served as the electron donor for ¹O₂ production. Chronoamperometric measurements showed a significant current increase upon PMS addition, and the open circuit potential (OCP) of the glassy carbon electrode coated with NFC/M rose immediately, indicating the strong surface interaction via electron transfer (Supplementary Fig. 34). However, phenol injection did not alter the current, signifying no electron interaction between phenol and PMS or the catalyst. A slight decrease in OCP could be attributed to the phenol adsorption on the NFC/M surface, possibly blocking active sites or modifying surface properties. Linear sweep voltammetry (LSV) analysis further confirms the electron transfer from PMS to the active sites, as evidenced by the increased current density at the NFC/M electrode (Supplementary Fig. 35)³⁰.

In in-situ EPR analysis, NVs were identified as key active sites. Compared to the stable unpaired electron signal on NFC without NVs (Supplementary Fig. 36), the presence of PMS led to a significant reduction in EPR signal intensity from NVs in NFC/M (Fig. 4i), indicating effective electron trapping. The signal initially decreased sharply and then stabilized, suggesting that NVs initially acted as electron acceptors to activate PMS, and subsequently as electron donors for ¹O₂ production, quickly reaching a dynamic equilibrium (Fig. 4i and Supplementary Fig. 37). These findings, aligning with our DFT simulations, provide robust experimental support for the mechanism of selective ¹O₂ production through NFC/M-activated PMS:

8. The authors announced that “the substitution of solvent H₂O with deuterium oxide (D₂O) to prolong ¹O₂ lifetime notably enhanced phenol degradation (line 226-227)”. However, it is really hard to say that we can find such a significant enhancement from Supplementary Fig. 26A (*k*_{obs} 0.581 min⁻¹ vs. 0.660 min⁻¹).

We acknowledge that the stated enhancement might appear less significant than implied and have rewritten our manuscript to reflect a more accurate interpretation.

In Fenton-like catalysis, D₂O is often used to extend the lifetime of ¹O₂ due to its ability to mitigate non-radiative deactivation processes compared to H₂O (Ren et al., *Environ. Sci. Technol.* 2022, 56, 78-97; Jensen et al., *J. Am. Chem. Soc.* 2010, 132, 8098-8105). The higher vibrational threshold of D₂O results in a slower rate of ¹O₂ deactivation, theoretically leading to enhanced reactive oxygen species (ROS) longevity and potentially more effective pollutant degradation (Park et al., *J. Am. Chem. Soc.* 2016, 138, 10734-10737; Liu et al., *Water Res.* 2021, 201, 117313).

However, D₂O can also affect the reactivity of the catalyst and the activation of the oxidant, often leading to contradictory effects (Takizawa et al., *Photochem. Photobiol. Sci.* 2015, 14, 1831-1843; Ogilby and Foote, *J. Am. Chem. Soc.* 1983, 105, 3423-3430). While it prolongs the lifetime of ¹O₂, it may also reduce the reactivity of the catalyst by altering the solvent dynamics and interaction energies, which can affect the overall activation of PMS and subsequent ROS production (Wu et al., *Proc. Natl.*

Acad. Sci. U.S.A. 2023, 120, e2305706120; Zhang et al., *Nat. Commun.* 2023, 14, 3538; Shao et al., *Environ. Sci. Technol.* 2021, 55, 16078-16087). These interactions might be applied to explain the modest increase observed in our degradation rates, reflecting the dual role of D₂O in influencing both ROS longevity and catalytic activity (Weng et al., *Angew. Chem. Int. Ed.* 2023, 135, e202310934; Liu et al., *Nat. Commun.* 2023, 14, 2881).

Our experiments aimed to utilize D₂O's effect to substantiate the involvement of ¹O₂ in the catalytic process. Although the increase in degradation rates from k_{obs} 0.581 min⁻¹ to 0.660 min⁻¹ was modest, it aligns with the expected impact of D₂O on ¹O₂ stability. This outcome reinforces the catalytic role of ¹O₂, as supported by our extensive EPR and mechanistic studies (Fig. 4 and Supplementary Figs. 23-30).

To better communicate these nuances, we have revised the relevant sections of the manuscript to clarify the implications of D₂O on our system and to ensure our descriptions accurately reflect the observed experimental outcomes (Supplementary Fig. 29).

Additionally, replacing water with deuterium oxide (D₂O) to extend the ¹O₂ lifespan moderately enhanced pollutant degradation, further conforming its key role in these reactions (Supplementary Fig. 29)¹. (P12, L225-228)

Notes for Supplementary Fig. 29

In Fenton-like catalysis, deuterium oxide (D₂O) was used to prolong the lifetime of ¹O₂ by reducing non-radiative deactivation processes that are more pronounced in H₂O. High-frequency O-H vibrations in H₂O deactivated ¹O₂ more effectively than the O-D vibrations in D₂O, leading to a significantly extended lifetime of ¹O₂ in D₂O (~67.0 μs) compared to H₂O (~3.5 μs). Such a higher vibrational threshold slows the rate of ¹O₂ deactivation, which could theoretically enhance the longevity of reactive oxygen species and improve pollutant degradation.

However, D₂O could also affect catalyst reactivity and oxidant activation, sometimes leading to contradictory effects. While extending the lifetime of ¹O₂, D₂O might simultaneously diminish catalyst reactivity by altering solvent dynamics and interaction energies. This modification can influence the overall activation of PMS and subsequent ROS generation. The modest increase in degradation rates observed in our experiments reflected the dual roles of D₂O in affecting both ROS longevity and catalytic activity.

9. Spelling mistakes: Line 115, “electrond”; Line 126, “phenol t on”; Line 309, “constructed” Line 311, “Tts”.

We apologize for the spelling errors and have corrected them (*electrons, phenol on, construct, Its*). Also, we have double-checked the main text and supplementary materials.

10. There are many manuscripts about exclusive singlet oxygen production in metal-free based Fenton-like system, what is your superiority expect for k_{obs} ?

Indeed, many studies have achieved ¹O₂ generation through various strategies, including the precise engineering of surface reaction sites and the optimization of

adsorbate configurations (Weng et al., *Angew. Chem. Int. Ed.* 2023, 62, e202310934; Xie et al., *Nat. Commun.* 2022, 13, 5560). Our work advances this field by not only concentrate on the production of $^1\text{O}_2$, but also its effective utilization, facilitated by our novel metal-free dual-site catalyst (NFC/M). This approach allowed us to simultaneously optimize two critical aspects of the Fenton-like reaction: the efficient and selective generation of reactive oxygen species (ROS), and their effective utilization.

- **Synergistic Dual-Site Mechanism:** Unlike typical metal-free catalysts, our NFC/M uniquely incorporated both nitrogen vacancies and F-C Lewis-acid sites. Such dual-site architecture facilitated simultaneous reaction and adsorption, enhancing the generation and effective utilization of $^1\text{O}_2$. The nitrogen vacancies efficiently captured terminal oxygen atoms from PMS, promoting $^1\text{O}_2$ formation, while the F-C sites effectively adsorbed pollutants, greatly reducing mass transfer limitations and increasing the local concentration of reactants.
- **Electron Configuration Control:** After precise modulation of electron distribution and surface polarization, our catalysts demonstrated an optimized electronic environment that promotes superior adsorption characteristics and reactivity, driven by the synergistic effects of long- and near-range interactions of nitrogen and fluorine. Such tailored electronic structure enhanced the selective activation of PMS and subsequent generation of $^1\text{O}_2$.
- **Advanced Material Characterization:** We employed comprehensive material characterization techniques, including in-situ Raman spectroscopy, in-situ EPR and electrochemical analyses, etc., to directly observe and quantify the dynamic interactions at the catalyst surface. These techniques provide a deeper understanding of the reaction mechanisms and surface interactions that are critical for effective catalyst design.
- **Environmental and Practical Applicability:** The catalyst's effectiveness in complex water matrices and industrial wastewater demonstrated its practical applicability and robustness. Our NFC/M was shown to be effective under real-world environmental conditions, addressing key challenges in water purification.
- **Sustainability and Safety:** Outstanding stability and reusability of the catalyst, as well as efficient utilization of oxidant, underscored its sustainability. Additionally, NFC/M was synthesized using montmorillonite-assisted pyrolysis, a process that ensured the absence of metal contaminants and reduced potential environmental and health hazards associated with metal-based catalysts.
- **Comprehensive Mechanistic Insights and Theoretical Underpinnings:** Our work provides new insights into the mechanisms of $^1\text{O}_2$ generation and utilization. DFT calculations helped clarify how the interactions between the catalyst's components enhanced overall efficacy, offering a comprehensive understanding of the reaction dynamics that govern the catalyst's behavior.

In summary, our work not only advances the field through enhanced reaction kinetics, but also achieve deep mechanistic insights via innovative catalyst design. In addition, we demonstrate its practical environmental applications and sustainability. These attributes collectively contribute to the superior performance and potential of our

catalyst system in real-world applications, distinguishing it from the existing technologies.

To address the reviewer's concern, we have provided more interpretation in the revised manuscript to elucidate the superiority of NFC/M-mediated Fenton-like catalysis (P18, L353-360).

This work not only underscores the rational design and controlled synthesis of a synergistic dual-site catalyst via electronic structure engineering but also enhances Fenton-like catalysis environmentally. It extends beyond improving reaction kinetics to include innovative catalyst design, deep mechanistic insights, and a focus on practical environmental applications and sustainability. These contributions signify a considerable advancement over the existing technologies, providing a solid foundation for developing next-generation catalysts for environmental remediation and beyond.

11. The authors need to check rearrangement & formatting carefully (e.g., the references list was irregularly formatted), please rearrange appropriately.

Accepting the reviewer's suggestions, we have carefully rearranged and reformatted the reference list and other sections of the manuscript. Thanks a lot!

Response to Reviewer 2's comments

Aiming at the problem of low utilization efficiency of singlet oxygen species ($^1\text{O}_2$) in water treatment, Gu et al. designed a transition metal-free NFC/M catalyst based on the guidance of theoretical calculation. The dual-site catalyst possessed nitrogen vacancies (NVs) and fluorine-carbon Lewis-acid sites, serving as reaction and adsorption centers, respectively. The authors proved that transition metal-free catalysts can achieve high catalytic activity guided by rational designing. The catalytic mechanism was elaborated as well. However, the manuscript contained several complexities that need be addressed before being considered for publication in Nature Communications.

We sincerely thank the reviewer for thoroughly examining our manuscript and providing very helpful comments to guide our revision.

1. The repeatability of NFC/M materials needs to verify. In the experimental section, it was stated that the catalyst was prepared by mixing several raw materials, and the vial was covered with a piece of aluminum foil with three fine holes. What is the purpose of this operation and how about the precision of this regulation? By this way, whether the content of F, and the N vacancy, can be precisely regulated? If possible, the correlation of their contents with the performances needs to be systematically studied.

The following is our response to the reviewer's questions:

(1) Repeatability verification:

To address the reviewer's concern, we have conducted additional control experiments to verify the repeatability of the NFC/M catalyst under the specified preparation conditions (detailed in the *Experimental Section*). These parallel experiments demonstrate good repeatability in the thermal synthesis of NFC/M, with minimal deviations observed in elemental composition (C/N ratio), nitrogen vacancy concentration (as indicated by similar EPR signal intensities), and performance in Fenton-like catalysis (phenol degradation) across different batches (Fig. R7). This result validates that our synthesis method was robust and repeatable.

Fig. R7 | Repeatability of NFC/M materials. **(a)** C/N molar ratio based on elemental analysis results. Error bars originate from statistical variations among materials synthesized in different batches. **(b)** Consistency of nitrogen vacancies evidenced by EPR spectra. **(c)** Performance consistency of parallel NFC/M samples in Fenton-like catalysis for pollutant degradation under controlled conditions. Conditions: [catalyst] = $0.2 \text{ g}\cdot\text{L}^{-1}$, [PMS] = 0.65 mM , [pollutant] = $20.0 \text{ mg}\cdot\text{L}^{-1}$, initial pH 7.0, Temp. = $20.0 \pm 2.0 \text{ }^\circ\text{C}$.

(2) Purpose and precision of the operation:

In our thermal synthesis process, the reaction vial was covered with aluminum foil pierced with three fine holes for the following objectives:

1) Controlled atmospheric exchange: The pierced aluminum foil allowed for limited gas exchange between the interior of the vial and the external environment, which is critical in controlling the atmosphere within the vial during high-temperature pyrolysis. This reaction involved the decomposition of melamine and polytetrafluoroethylene (PTFE) precursors, releasing gases. Providing a relatively closed space facilitated the mixing and retention of carbon, nitrogen, and fluorine precursors.

2) Release of gaseous product: During heating, it was essential to timely release water vapor and other gaseous by-products like SiF₄, which are instrumental in the in-situ removal of the MMT template.

3) Prevention of contamination: The foil acted as a barrier, preventing any particles or environmental contaminants from entering the vial during the heating process. This is especially important for maintaining the purity and consistency of the reaction conditions.

To ensure the precision and reproducibility of this setup, the aluminum foil was systematically pierced with a standard 5.0-ml syringe needle. This method has been successfully applied by other research groups because of its reliability and reproducible sample preparation (Xu et al., *Nat. Sustain.* 2021, 4, 233-241). This precise operation guaranteed consistent atmospheric conditions and gas exchange rates during thermal synthesis, minimizing batch-to-batch variations in the synthesis of NFC/M materials.

(3) Content regulation:

Fig. R8 | The regulation of NFC/M materials prepared with various fluorine precursors. **(a)** Remaining SiO_x in the different NFC/M samples, as indicated by EPR results. **(b)** Variation in nitrogen vacancies across NFC/M samples, evidenced by EPR spectra. **(c)** Comparative analysis of carbon, nitrogen, and nitrogen vacancies in different NFC/M samples. **(d)** Fluorine content variation in NFC/M samples.

The precise control over the concentration of nitrogen vacancies (NVs) and fluorine (F) in NFC/M was inherently challenging due to their derivation from distinct precursors—melamine and PTFE, respectively (refer to the *Experimental Section* for details). Nevertheless, the interplay between F doping and the formation of defective NVs allowed us to achieve partial regulation of both NV concentration and F content by adjusting the amount of the fluorine precursor used in the synthesis process.

To examine this modulation potential, we have conducted a series of controlled experiments by varying the fluorine precursor dosage: 0.0, 0.4, 0.8, 1.2, 1.6, and 2.0 mL (**Fig. R8**). EPR results show that, as the dosage of PTFE increased, the degree of MMT template removal intensified and the concentration of NVs initially increased, but then declined (**Fig. R8a, b**). Additionally, elemental analysis (EA) and X-ray photoelectron spectroscopy (XPS) indicate that, at higher PTFE dosages, there was a corresponding increase in carbon and fluorine contents in the materials, while the nitrogen content exhibited a slight but stable decrease (**Fig. R8c, d**).

These findings suggest that varying the fluorine precursor dosage allowed partial control over the carbon/nitrogen ratio and the concentration of nitrogen vacancies in NFC/M. This capability to modulate these variables is crucial for optimizing the catalyst's performance in Fenton-like catalysis by tailoring the co-pyrolyzed fluorine precursor's contribution to the synthesis process.

(4) Correlation of their contents with the performances:

Fig. R9 | The regulation of NFC/M materials prepared with various fluorine precursors for Fenton-like catalysis. **(a)** Comparison of pollutant removal by different NFC/M samples through PMS activation. **(b)** Correlation of nitrogen vacancies with Fenton-like reactivity in various NFC/M samples, analyzed through EPR spectra and pollutant degradation tests. Testing conditions: [catalyst] = 0.2 g·L⁻¹, [PMS] = 0.65 mM, [pollutant] = 20.0 mg·L⁻¹, initial pH 7.0, Temp. = 20.0 ± 2.0 °C.

Fig. R10 | The regulation of electronic structure of NFC/M materials prepared with various fluorine precursors. **(a)** Comparison in the binding energy of nitrogen in different NFC/M samples based on XPS spectra. **(b)** Comparison in the binding energy of carbon in different NFC/M samples based on XPS spectra. **(c)** Comparison in the binding energy of fluorine in different NFC/M samples based on XPS spectra.

The performance of the NFC/M catalyst was intricately linked to the concentrations of NVs and F, which greatly influenced the catalyst's surface electronic

structure (**Fig. R8-10**). Our study demonstrates that the long-range interactions of fluorine and near-range interactions of nitrogen synergistically modulated the electron distribution and polarization within the carbon matrix. This results in asymmetric electron configurations and electron-deficient nitrogen vacancies, which are crucial for optimizing PMS activation and pollutant degradation (Figs. 1, 3 and 5).

Catalytic activity in the NFC/M correlated with the changes in NV concentration, exhibiting an initial increase followed by a decrease, with the optimal performance observed at an intermediate fluorine content (F-1.2) (**Fig. R9**). However, the relationship between the catalytic performance and NV concentration was not strictly linear, as it was heavily influenced by the fluorine content and the resultant alterations in the electronic structure. XPS analysis supports this conclusion, showing significant shifts in the binding energies of C, N, and F that corresponded to the variations in NV and F concentrations (**Fig. R10**). These changes suggest dynamic adjustments in the catalyst's surface electronic structure, impacting both PMS activation and pollutant degradation.

The presence of F-C Lewis acid sites further modifies the atomic and electronic structures, enhancing the catalyst's reactivity and surface interaction capabilities. Such a complex interplay between structural factors and catalytic performance was further validated by the consistent trends in Fenton-like reactivity observed with varying NV concentrations across different dosages of PTFE during thermal preparation. Despite these consistent trends, no strict linear relationship existed between Fenton-like reactivity and NV concentration due to the regulatory effects of F-C Lewis acid sites on the atomic and electronic structures and surface enrichment (**Figs. R8-R10**).

These detailed insights underscore the nuanced control of electronic structure through the interaction of fluorine and nitrogen, highlighting the complex relationship between structural factors and enhanced catalytic performance for water purification. This comprehensive understanding enables precise tailoring of catalytic systems to effectively address specific environmental challenges.

To address the reviewer's concern, we have added these experimental results and their interpretation into the revised manuscript (**Supplementary Note 2; Supplementary Figs. 42-45**).

Deeper insights into the catalytic activity were obtained by assessing the impacts of NVs and F content on NFC/M (Supplementary Figs. 42-45). The catalyst's performance was closely linked to these concentrations, which crucially influenced its surface electronic structure. Specifically, the catalytic activity correlated with the variations in NV concentration, initially increasing and later diminishing (Supplementary Figs. 43 and 44). This observation supports our catalyst design considerations that the electronic structure engineering of NV drastically boosted PMS activation, resulting in excellent performance facilitated by the synergistic effects of dual-site interactions. (P15, L287-295)

The vial was then sealed with aluminum foil punctured with three fine holes and heated to 550 °C at a rate of 13.0 °C/min in a muffle furnace for 1 h (Supplementary Note 2). The fluorine and nitrogen vacancy content in NFC/M could be adjusted by varying the amount of PTFE in the precursor. (P19, L385-388)

Supplementary Note 2. Thermal operation

In our thermal synthesis process, several crucial objectives were achieved by covering the reaction vial with aluminum foil pierced with three fine holes⁷. These objectives were: 1) Controlled atmospheric exchange: The pierced aluminum foil allowed for limited gas exchange between the interior of the vial and the external environment, which was critical in controlling the atmosphere within the vial during high-temperature pyrolysis. This reaction involved the decomposition of melamine and PTFE organic precursors, releasing gases. Providing a relatively closed space facilitated the mixing and retention of carbon, nitrogen, and fluorine precursors; 2) Release of gaseous product: During heating, it was essential to timely release water vapor and other gaseous by-products like SiF₄, which were instrumental in the in-situ removal of the MMT template; and 3) Prevention of contamination: The foil acted as a barrier, preventing any particles or environmental contaminants from entering the vial during the heating process. It was especially important to maintain the purity and consistency of the reaction conditions.

To ensure the precision and reproducibility of this setup, the aluminum foil was systematically pierced with a standard 5.0-ml syringe needle. This precise operation guaranteed consistent atmospheric conditions and gas exchange rates during thermal synthesis, minimizing batch-to-batch variations in the synthesis of NFC/M materials.

Supplementary References

7. Xu, J. et al. Organic wastewater treatment by a single-atom catalyst and electrolytically produced H₂O₂. Nat. Sustain. 4, 233-241 (2021).

Notes for Supplementary Fig. 42

These parallel experiments demonstrate good repeatability in the thermal synthesis of NFC/M, with minimal deviations observed in elemental composition (C/N ratio), nitrogen vacancy concentration (as indicated by similar EPR signal intensities), and performance in Fenton-like catalysis (phenol degradation) across different batches (Supplementary Fig. 42). This result indicates that our synthesis method was robust and repeatable.

Notes for Supplementary Fig. 43

EPR results demonstrate that, as the dosage of PTFE increased, the degree of MMT template removal intensified and the concentration of NVs initially increased, but then declined (Supplementary Fig. 43a, b). Additionally, EA and XPS indicate that, with higher PTFE dosages, there was a corresponding increase in carbon and fluorine content in the materials, while the nitrogen content exhibited a slight but stable decrease (Supplementary Fig. 43c, d).

These findings suggest that varying the fluorine precursor dosage allowed partial control over the carbon/nitrogen ratio and the concentration of nitrogen vacancies in NFC/M. The capability to modulate these variables was crucial for optimizing the catalyst's performance in Fenton-like catalysis by tailoring the co-pyrolyzed fluorine

precursor's contribution to the synthesis process.

Notes for Supplementary Fig. 44

The Fenton-like reactivity of NFC/M for pollutant degradation closely followed the concentration changes of defective nitrogen vacancies resulting from varying PTFE dosages during thermal preparation. While there was a greatly positive correlation, it was not strictly linear; mainly because both the PMS activation and pollutant degradation processes were influenced by the presence of F-C Lewis acid sites. These sites contributed to the regulated atomic and electronic structures and enhanced surface interactions, complicating the direct correlation between nitrogen vacancy concentration and catalytic performance.

Notes for Supplementary Fig. 45

XPS analysis shows significant shifts in the binding energies of C, N, and F, which corresponded to the variations in NV and F concentrations (Supplementary Figs. 43, 45). These changes suggest dynamic adjustments in the catalyst's surface electronic structure, affecting both PMS activation and pollutant degradation (Supplementary Fig. 44).

2. Elements of the template, such as Al, Si and Na, were remained in the synthesized material. How many contents of these elements in the catalyst? Does the presence of Al, Si or Na affects the catalytic performance? How many C, N, F contents present in the catalyst?

In our study, trace amounts of Al, Si, and Na from the montmorillonite (MMT) template were indeed retained in the synthesized material. Specifically, the atomic percentages of Al, Si, and Na were quantified as 1.16%, 0.34%, and 1.24% respectively, as determined by XPS analysis (**Table R2**). The contents of C, N, and F in the catalyst were measured at 63.45%, 24.14%, and 4.17%, respectively.

Regarding the performance, our analyses indicate that the residual Al, Si, and Na were not involved in the catalytic process (**Fig. R11**). XPS analysis confirms that these elements existed in inert forms such as montmorillonite ($\text{SiO}_2/\text{Al}_2\text{O}_3$), Na^+ , and minor AlF_3 (**Figs. R12-R15**), and remained structurally uninvolved in the catalytic process. This inert nature was corroborated by the control experiments with NC/M and FC/M samples, which had similar or higher contents of these elements but exhibited negligible activity under identical Fenton-like conditions (**Fig. 5b and Fig. R11A**). Further experiments involving the removal of residual Si, Na, and Al_2O_3 from NFC/M using a 4-h treatment with a 1.0 M HF solution demonstrate that the catalytic activity remained stable post-etching (**Fig. R11B**). This result confirms that the presence of these template-derived elements did not affect the Fenton-like reactivity of NFC/M. The superior performance of NFC/M was primarily attributed to the nitrogen vacancies and the F-C Lewis acid sites, which effectively regulated the atomic and electronic structures for the optimized PMS activation and pollutant degradation.

Table R2 | Elemental contents of C, N, F, Si, Al and O based on XPS analysis for NFC/M, NC/M, FC/M and NFC/M-HF.

Samples		C 1s	N 1s	F 1s	Si 2p	Al 2p	O 1s	Na 1s
NFC/M	BE (eV)	284.8	398.48	686.68	103.18	74.48	532.25	1071.44
	at%	63.45	24.14	4.17	0.34	1.16	5.5	1.24
	wt%	58.03	25.01	6.05	0.62	2.31	6.05	1.93
NC/M	BE (eV)	288.09	398.59	---	102.91	74.74	532.25	1071.63
	at%	25.65	18.10	---	12.01	3.75	39.4	1.09
	wt%	18.41	15.34	---	20.06	6.16	38.22	1.81
FC/M	BE (eV)	284.8	---	687.61	103.79	77.72	532.25	1073.52
	at%	65.08	---	23.37	0.38	1.99	8.07	1.11
	wt%	59.62	---	26.86	0.69	3.86	7.23	1.74
NFC/M-HF	BE (eV)	284.8	398.48	686.68	103.18	77.68	532.25	1071.44
	at%	61.58	26.57	4.87	0.06	1.38	5.49	0.05
	wt%	56.83	26.73	6.97	0.11	2.75	6.43	0.08

Fig. R11 | The impacts of the residual Si, Al, and Na from MMT template on the NFC/M-mediated Fenton-like catalysis for PMS activation and pollutant degradation. **(a)** Comparison in pollutant removal by NFC/M and FC/M. **(b)** Comparison in pollutant removal by NFC/M before and after HF etching for Si/Al/Na corrosion. Testing conditions: [catalyst] = 0.2 g·L⁻¹, [PMS] = 0.65 mM, [pollutant] = 20.0 mg·L⁻¹, initial pH 7.0 (if not adjusted), Temp. = 20.0 ± 2.0 °C.

To address the reviewer’s concern, we have added the above experimental results and their interpretation into the revised manuscript (Supplementary Methods; Supplementary Fig. 41; Supplementary Table 3).

During synthesis, MMT acted as an in-situ hard template, and was subsequently etched away by F-rich gas released from the decomposition of polytetrafluoroethylene (PTFE) (Supplementary Fig. 11 and Supplementary Table 3)³⁴. (P8, L138-141)

Abundant F-C Lewis acid sites were identified in NFC/M with a molar ratio of F reaching up to 4.2% (Fig. 3e and Supplementary Fig. 15f and Supplementary Table 3)³⁴. (P10, L174-175)

Such notable disparity in reactivity also underscores that residual Si, Al, and Na within the catalyst did not contribute to its catalytic activity (Supplementary Fig. 41). (P15, L285-287)

Preparation of control sample NFC/M-HF

To exclude the possible impact of residual Si, Al, and Na on the catalytic performance of NFC/M, the synthesized material underwent an acid-washing procedure. Specifically, 500 mg of NFC/M was added to 60 mL of 1 M hydrofluoric acid (HF) and ultrasonicated for 3 min. This was followed by continuous magnetic stirring for 4 h to ensure thorough acid washing (Supplementary Table 3). The mixture was then vacuum filtered and washed eight times with deionized water until the effluent pH became neutral. Finally, the washed catalyst was dried at 60 °C for 12 h. The resulting material was designated as NFC/M-HF.

3. The authors characterized the F-C Lewis locus and emphasized its critical role in the reaction. However, according to the preparation method, is it possible that the F elements migrate to Al and Si substrate? As Al₂O₃ and SiO₂ are very suitable for the location of active species. Also, is it possible that the F elements bind to Na due to the electronegativity?

We present the following response to the reviewer's insightful questions:

- 1) F interaction with Si and Al substrates

We highly agree with the reviewer's comment that Al₂O₃ and SiO₂ are very suitable for the structural location of active species (Wang et al., *ACS Catal.* 2022, 12, 2632-2638; Jiao et al., *Nat. Commun.* 2020, 11, 2831). Our MMT-mediated co-pyrolysis for NFC/M preparation was proposed and validated based largely on these features.

In the process of synthesizing NFC/M, the montmorillonite (MMT) template, composed of a SiO₂-sandwiching-Al₂O₃ layered structure, played a crucial role (**Figs. 2, 3 and R12**). During thermal preparation, the highly reactive fluorine intermediates, derived from the decomposition of PTFE, interacted preferentially with the outer SiO₂ layers of MMT. This interaction led to the formation of SiF₄, facilitating the in-situ removal of SiO₂ (SiO₂ + 4HF → SiF₄↑ + 2H₂O). Subsequently, the inner Al₂O₃ layers provided a buffering effect, consuming any excess corrosive fluorine to form AlF₃, which was also confirmed by the minimal residual presence of Si in XPS analysis (**Fig. R12**) and the formation of some AlF₃ in the catalyst matrix (**Figs. R13 and R14**). This staged consumption effectively prevented the corrosion of the carbon structure by acidic fluorine gas (Supplementary Fig. 16).

Fig. R12 | Comparison of the residual SiO₂ contents in the different catalyst samples of NC/M, NFC/M, NFC/M-HF and FC/M prepared under various conditions and the atomic representation of used layer-stacked MMT template.

Fig. R13 | Comparison of the binding energy and chemical speciation of aluminum in NC/M, NFC/M, NFC/M-HF and FC/M based on XPS spectra.

Fig. R14 | Comparison of the binding energy and chemical speciation of aluminum in different NFC/M catalysts prepared with various fluorine precursors based on XPS spectra.

Fig. R15 | Comparison of the binding energy and chemical speciation of sodium. **(a)** For NC/M, NFC/M, NFC/M-HF and FC/M based on XPS spectra. **(b)** For different NFC/M catalysts prepared with various fluorine precursors based on XPS spectra.

2) Potential binding of F to Na:

As for the interaction of F with Na, our XPS analysis indicates that there was no significant formation of NaF under standard conditions (NFC/M; **Fig. R15a**). Such absence could be attributed to the reductive atmosphere created by the decomposition of melamine, which facilitated the formation of HF, rather than NaF. HF was then consumed or buffered by the MMT template (Fig.2 and **Figs. R12-R14**). However, in the samples synthesized with higher dosages of PTFE, or without melamine, we observed a gradual formation of NaF (**Fig. R15**). This result suggests that in environments lacking the reductive atmosphere, excess fluorine was free to react with available Na, forming NaF, further highlighting the selectivity of fluorine binding under varying synthetic conditions.

These observations indicate that, while SiO₂ and Al₂O₃ in the MMT were initially reactive towards corrosive fluorine, leading to their consumption and the stabilization of the carbon matrix, the residual fluorine did not largely bind to Na under controlled synthetic conditions. Moreover, the catalytic activity of our NFC/M catalyst was primarily driven by the nitrogen vacancies and F-C Lewis acid sites, which regulated

the atomic and electronic structures essential for effective Fenton-like catalysis, rather than by any residual Al, Si, or Na from the MMT template (**Fig. R11**). Thus, the migration of fluorine to Al and Si substrates and its interaction with Na did not adversely impact the desired catalytic properties of NFC/M.

To address the reviewer's concern, we have added the above experimental results and their interpretation into the revised manuscript (**Supplementary Figs. 18-20**).

These findings highlight the protective role of MMT during the synthesis of NFC/M. The decomposition of melamine created a reductive atmosphere, facilitating the formation of corrosive byproducts like HF (Supplementary Fig. 18), which was sequentially consumed or buffered by the SiO₂ and Al₂O₃ layers within the MMT template (Supplementary Figs. 19 and 20), effectively mitigating the corrosive byproducts from PTFE decomposition. Such a protective mechanism shields the carbon matrix from chemical etching, thus promoting the selective formation of pyridinic and pyrrolic NVs³⁴. This interplay underscores the importance of MMT in safeguarding the structure of the catalyst during synthesis, ensuring the stability and functionality of the resulting NFC/M. (P9, L164-173)

Notes for Supplementary Fig. 18

XPS analysis indicates that there was no significant formation of NaF under standard conditions (Supplementary Fig. 18a). This absence could be attributed to the reductive atmosphere created by the decomposition of melamine, which facilitated the formation of HF, rather than NaF. HF was then consumed or buffered by the MMT template (Supplementary Figs. 19 and 20). However, in the samples synthesized with higher dosages of PTFE, or without melamine, we observed a gradual increase in NaF formation (Supplementary Fig. 18). This result suggests that in environments lacking the reductive atmosphere, excess fluorine was free to react with available Na, forming NaF, further highlighting the selectivity of fluorine binding under varying synthetic conditions.

Notes for Supplementary Figs. 19 and 20

In the process of synthesizing NFC/M, the montmorillonite (MMT) template, composed of a SiO₂-sandwiching-Al₂O₃ layered structure, played a crucial role (Figs. 2, 3 and Supplementary Figs. 19). During thermal preparation, the highly reactive fluorine intermediates, derived from the decomposition of polytetrafluoroethylene (PTFE), interacted preferentially with the outer SiO₂ layers of MMT. This interaction led to the formation of SiF₄, facilitating the in-situ removal of SiO₂ ($\text{SiO}_2 + 4\text{HF} \rightarrow \text{SiF}_4\uparrow + 2\text{H}_2\text{O}$). Subsequently, the inner Al₂O₃ layers provided a buffering effect, consuming any excess corrosive fluorine to form AlF₃, which was also confirmed by a minimal residual presence of Si in XPS analysis (Supplementary Figs. 19a) and the formation of some AlF₃ in the catalyst matrix (Supplementary Figs. 20). This staged consumption effectively prevented the corrosion of the carbon structure by acidic fluorine gases (Supplementary Fig. 16).

4. The effects of concentrations of the quenching agents (TBA/methanol/ethanol) on

the pollutant degradation activity should be studied and discussed.

Accepting the reviewer's suggestion, we have conducted additional experiments using the radical-quenching agents like methanol (MeOH), ethanol (EtOH), and tert-butanol (TBA) with varied concentrations (0.001-0.43 M) to explore their effects on pollutant degradation in the NFC/M-mediated Fenton-like system (**Fig. R16**).

Fig. R16 | Effects of ROS scavengers on phenol degradation in NFC/M system. **(a)** Concentration effects of MeOH scavenger. **(b)** Concentration effects of EtOH scavenger. **(c)** Concentration effects of TBA scavenger. **(d)** Comparative effects of different scavengers on the reaction kinetics. Reaction conditions: [catalyst] = $0.2 \text{ g} \cdot \text{L}^{-1}$, [PMS] = 0.65 mM , [pollutant] = $20.0 \text{ mg} \cdot \text{L}^{-1}$, initial pH 7.0, Temp. = $20.0 \pm 2.0 \text{ }^\circ\text{C}$.

As shown in **Fig. R16a-c**, neither low nor excessive concentrations of MeOH, EtOH, or TBA greatly impeded phenol degradation, which aligns with previous findings. The minimal inhibition observed at high TBA concentrations could be attributed primarily to the increase in the viscosity of the reaction solution, which impedes mass transfer (Abdülvahap and Özcan, *Fuel* 2022, 315, 123200; Dai et al., *Sep. Purif. Technol.* 2021, 279, 119717; Kim et al., *Nat. Commun.* 2020, 11, 5146). Conversely, the three specific $^1\text{O}_2$ -quenching agents—TEMP, NaN_3 , and FFA—consistently played a dominant role across all tested concentrations (0.4 to 40.0 mM) (**Fig. R16d**).

The control results kept highly consistent with our EPR measurements as well as the measured water matrix effects under different testing conditions (Figs. 4e and 5d, i). These results not only validate our previous insights into the mechanisms of Fenton-like catalysis (Fig. 4d and Supplementary Fig. 28), but also reinforce the exclusive

generation and utilization of $^1\text{O}_2$ in pollutant degradation facilitated by NFC/M, where hydroxyl and sulfate radicals showed minimal participation (Fig. 4).

To address the reviewer's concern, we have added the above experimental results and their interpretation into the revised manuscript (Fig. 4d; Supplementary Fig. 28).

Subsequent scavenging experimental results highlight the pivotal role of $^1\text{O}_2$ in pollutant degradation (Fig. 4d). The introduction of radical-quenching agents like methanol (MeOH), ethanol (EtOH), and tert-butanol (TBA) barely influenced phenol and bisphenol A (BPA) degradation, suggesting the negligible involvement of the other reactive species (Supplementary Fig. 28a-f)^{1,5}. Conversely, specific $^1\text{O}_2$ -quenching agents, TEMP, NaN_3 and furfuryl alcohol (FFA), at any concentration drastically inhibited pollutant degradation with the effect intensifying concomitant with the scavengers' concentration, highlighting the dominant role of $^1\text{O}_2$ in the NFC/M-mediated Fenton-like catalysis (Supplementary Fig. 28g-i). (P11-12, L217-225)

Notes for Supplementary Fig. 28

Dosing MeOH and EtOH (scavengers for $\text{SO}_4^{\cdot-}$ and $\cdot\text{OH}$) or TBA (scavenger for $\cdot\text{OH}$) (1.0 mM to 0.43 M) exhibited minimal inhibitory effects on phenol degradation. This result indicates the absence of $\cdot\text{OH}$ and $\text{SO}_4^{\cdot-}$ as active species. The very slight inhibition of excessive TBA might primarily be attributed to the increased solution viscosity, which impedes mass transfer.

5. The author should give more deeply discussion towards the identification of the free radicals, especially the hydroxyl radicals and the sulfate radicals, and the possibility of electron transfer during the process.

Accepting the reviewer's suggestion, we have provided additional evidence and expanded our discussion on the identification of free radicals, specifically hydroxyl and sulfate radicals, and explored the potential for electron transfer during the process in the revised manuscript.

In our work, a comprehensive suite of experimental techniques was utilized, including EPR, radical quenching tests, fluorescent probe tests, electrochemical analyses, and Raman spectroscopy, to rigorously identify the reactive species during the Fenton-like reactions facilitated by our NFC/M catalyst (Fig. 4 and Supplementary Fig. 23-33). When using 5,5-dimethyl-1-pyrroline-N-oxide (DMPO) to trap potential radicals, no characteristic EPR signals indicative of $\cdot\text{OH}$, $\text{SO}_4^{\cdot-}$, and $\text{O}_2^{\cdot-}$ were observed (Fig. 4e and Supplementary Fig. 23, 32). Instead, a DMPOX signal, indicative of $^1\text{O}_2$, was consistently detected, as confirmed by the NaN_3 probe (Gao, et al., *Angew. Chem. Int. Ed.* 2021, 60, 22513-22521), underscoring the absence of typical radical species observed in traditional Fenton reactions.

Furthermore, radical quenching tests showed negligible impact of typical $\cdot\text{OH}$ and $\text{SO}_4^{\cdot-}$ scavengers (TBA, MeOH, EtOH) on pollutant degradation, reinforcing the minimal role of these radicals in our system (Supplementary Fig. 28a-d). Conversely, the introduction of known $^1\text{O}_2$ scavengers (TEMP, NaN_3 , FFA) at varying concentrations largely inhibited the degradation process, clearly highlighting the predominant role of $^1\text{O}_2$ in the catalysis (Supplementary Fig. 28g-i). The enhancement

of phenol degradation upon substituting H₂O with D₂O, which extends the lifespan of ¹O₂ (Weng et al., *Angew. Chem. Int. Ed.* 2023, 135, e202310934), further supports this finding (Supplementary Fig. 29).

To investigate the possible non-radical electron transfer, we utilized a typical galvanic oxidation system (GOS) with both blank and NFC/M-modified electrodes (Huang et al., *Environ. Sci. Technol.* 2019, 53, 12610-12620) (Supplementary Fig. 30a). The absence of phenol degradation in these systems suggests negligible catalyst-mediated electron transfer processes (Yin et al., *Appl. Catal. B* 2023, 338, 123029) (Supplementary Fig. 30b). This is further corroborated by chronoamperometric measurements where dosing phenol did not affect the current either in the presence or absence of PMS, indicating no electron interaction between phenol and PMS or the catalyst (Supplementary Fig. 34). Additionally, the presence of phenol had a minimal impact on PMS decomposition (Supplementary Fig. 24), and in-situ Raman spectra did not reveal any reactive surface complexes (PMS*), effectively ruling out direct oxidation processes mediated by the catalyst-PMS complex (Duan et al., *Appl. Catal. B* 2021, 299, 120714) (Fig. 4h).

The above comprehensive assessments conclusively demonstrate that ¹O₂, rather than hydroxyl or sulfate radicals, was the principal reactive species in our NFC/M-mediated Fenton-like system, indicating that neither radical formation nor electron transfer contributed to the observed pollutant degradation significantly.

To address the reviewer's concern, we have added the above experimental results and their interpretation into the revised manuscript (Fig. 4d; Supplementary Figs. 23, 24, 28).

When employing 5,5-dimethyl-1-pyrroline-N-oxide (DMPO) to trap potential radicals, no characteristic EPR signals indicative of [•]OH, SO₄^{•-}, and O₂^{•-} were observed, but only the ¹O₂-mediated DMPOX signal was detected, as confirmed by the NaN₃ probe (Supplementary Fig. 23b-d)⁵, further corroborating the selective generation of ¹O₂. (P11, L209-213)

The introduction of radical-quenching agents like methanol (MeOH), ethanol (EtOH), and tert-butanol (TBA) barely influenced phenol and bisphenol A (BPA) degradation, suggesting the negligible involvement of the other reactive species (Supplementary Fig. 28a-f)^{1,5}. (P12, L218-221)

No pollutant removal was observed in the galvanic oxidation system (GOS) with either blank or NFC/M-modified carbon electrodes, indicating negligible catalyst-mediated electron transfer from pollutant to oxidant (Supplementary Fig. 30)⁴³. (P12, L228-230)

However, phenol injection did not alter the current, signifying no electron interaction between phenol and PMS or the catalyst. (P13, L253-255)

Notes for Supplementary Fig. 23

When DMPO was utilized to detect radicals, no discernible signals of [•]OH and SO₄^{•-} radicals were observed, only a DMPOX signal was detected. To examine the origin of DMPOX signal, EPR quenching experiments were conducted (Supplementary Fig. 23). These experiments demonstrate that typical radical quenchers did not affect

the DMPOX signal. However, NaN_3 , a known $^1\text{O}_2$ quencher, greatly inhibited the formation of the DMPOX signal. In addition, the presence of NaN_3 showed minimal impact on the consumption of PMS (Supplementary Fig. 24). Thus, the DMPOX signal was originated from the DMPO oxidation by $^1\text{O}_2$.

Notes for Supplementary Fig. 24

The presence of phenol had a minimal promoting effect on PMS decomposition (Supplementary Fig. 24), and in-situ Raman spectra did not reveal any reactive surface complexes (PMS^* ; Fig. 4h), effectively ruling out the direct oxidation processes mediated by the catalyst-PMS complex.

Notes for Supplementary Fig. 28

Dosing MeOH and EtOH (scavengers for $\text{SO}_4^{\cdot-}$ and $^{\cdot}\text{OH}$) or TBA (scavenger for $^{\cdot}\text{OH}$) (1.0 mM to 0.43 M) exhibited minimal inhibitory effects on phenol degradation. This result indicates the absence of $^{\cdot}\text{OH}$ and $\text{SO}_4^{\cdot-}$ as active species. The very slight inhibition of excessive TBA might primarily be attributed to the increased solution viscosity, which impedes mass transfer.

- In Fig 3c, the author used SEM-EDS to obtain the C/N ratio of the catalyst. More reliable data are suggested to provide.

Accepting the reviewer's suggestion, we have added more reliable data on the C/N ratio, additional measures beyond the initial SEM-EDS and XPS analyses into the revised manuscript (Supplementary Fig. 16a, b and Supplementary Table 3). In line with the established methods (Su et al., *Angew. Chem. Int. Ed.* 2021, 133, 21431-21436; Rasam et al., *Fuel* 2020, 280, 118665; Kumar et al., *J. Mater. Chem. A*, 2021, 9, 111), we have utilized elemental analysis (EA) to accurately quantify the C/N ratio. This method involves complete combustion of the samples, resulting in the formation of CO_2 and N_2 , from which the C and N contents were precisely determined (Fig. R17).

Fig. R17 | C/N molar ratio in NFC, NC/M and NFC/M catalysts based on elemental analysis (EA). The molar ratio of carbon to nitrogen (C/N) in each catalyst was determined through elemental analysis using an Elementar Vario EL cube instrument (Elementar, Germany). The combustion tube was heated to 950 degrees Celsius, and the reduction tube to 550 degrees Celsius. Error bars reflect statistical variations among parallel samples synthesized in different batches.

The elemental ratios determined via EA in NFC, NC/M, and NFC/M were consistent with those obtained from surface analysis via SEM-EDS and XPS (Fig. 3c and Supplementary Fig. 16a, b). This cross-validation of data with various techniques confirms the accuracy of our findings. The results highlight the role of the MMT template in facilitating formation of specific vacancies and F-C Lewis acid sites (Figs. 2, 3 and Supplementary Figs. 15-17).

To address the reviewer's concern, we have expanded the experimental section to include these new results and provided a comprehensive interpretation of the findings, ensuring a more robust validation of the C/N ratios and their implications for the catalyst's functionality (Fig. 3c; Supplementary Fig. 16).

In addition, the increased bulk phase C/N atomic ratio of 1.47 (vs. 0.60 for NC/M and 0.91 for NFC) reaffirmed the presence of NVs in NFC/M (Fig. 3c), aligning closely with the surface measurements (Supplementary Fig. 16a, b). (P9, L157-160)

The carbon to nitrogen (C/N) molar ratio in the catalyst samples was determined via elemental analysis (EA) with an instrument (Vario EL cube, Elementar Co., Germany), with the combustion tube heated to 950 °C and the reduction tube at 550 °C. (P20, L409-412)

7. The abstract is too long. The author is suggested to refine the language.

Accepting the reviewer's suggestion, we have rewritten the abstract as follows:

Developing eco-friendly catalysts for effective water purification with minimal oxidant use is imperative. Herein, we present a novel metal-free and nitrogen/fluorine dual-site catalyst, enhancing the selectivity and utilization of singlet oxygen (1O_2) for water decontamination. Advanced theoretical simulations reveal that synergistic fluorine-nitrogen interactions modulate electron distribution and polarization, creating asymmetric surface electron configurations and electron-deficient nitrogen vacancies. These properties trigger the selective generation of 1O_2 from peroxymonosulfate (PMS) and improve the neighboring utilization of reactive oxygen species, facilitated by contaminant enrichment at the fluorine-carbon Lewis-acid adsorption sites. Utilizing these insights, we synthesize the catalyst through montmorillonite (MMT)-assisted pyrolysis (NFC/M). This method leverages the role of MMT as an in-situ layer-stacked template, enabling controlled decomposition of carbon, nitrogen, and fluorine precursors and resulting in a catalyst with enhanced structural adaptability, reactive site accessibility, and mass-transfer capacity. The NFC/M demonstrates an impressive 290.5-fold increase in phenol degradation efficiency than the single-site analogs, outperforming most of metal-based catalysts. This work not only underscores the potential of precise electronic and structural manipulations in catalyst design, but also advances the development of efficient and sustainable solutions for water purification. (P2, L1-19)

REVIEWERS' COMMENTS

Reviewer #1 (Remarks to the Author):

The authors have revised the manuscript according to the comments and suggestions. I think this manuscript can be published in this journal.

Reviewer #2 (Remarks to the Author):

I have thoroughly read the revised manuscript and thought that the manuscript was well improved and could be accepted for publication. The authors have well answered the questions of the reviewers.